

# Sorting sudden stratospheric warmings with the downward tropospheric influence using ERA5 and CESM2-WACCM

Rongzhao Lu[1] and Jian Rao[1*]

[1]Collaborative Innovation Center on Forecast and Evaluation of Meteorological Disasters / Key Laboratory of Meteorological Disaster of Ministry of Education, Nanjing University of Information
Science and Technology, Nanjing 210044, China

*Correspondence to*: Jian Rao (raojian@nuist.edu.cn)

**Abstract.** Sudden Stratospheric Warming events (SSWs) can have a downward impact on the troposphere, but the mechanism remains uncertain. To improve the understanding of this uncertainty, this study explores the characteristics of SSWs that have different impacts on the troposphere. Using the
ERA5 data and CESM2-WACCM outputs, 52 SSWs are identified in ERA5 and 273 in CESM2-WACCM, with 33 and 119 downward-propagating SSWs (DWs), respectively. The DWs are classified into three types based on cold surges over Eurasia (EA), North America (NA), and both (BOTH), respectively. Both DWs and non-downward-propagating SSWs (NDWs) correspond to weakened and/or deformed the polar vortex, but DWs induce stronger negative Northern Annular Mode (NAM) and North
Atlantic Oscillation (NAO) responses. For DWs, the anomalous high develops in the polar region, which deflects to lower latitudes, consistent with the frequent appearance of the polar high and the midlatitude blockings. The shape of the anomalous polar high varies with the DW type, and the extension and shift of the anomalous high lead to different surface responses. The DWs are also accompanied by a southward shift of the precipitation belt, especially over the oceanic and coastal regions. NDWs show relatively
weaker impact on the troposphere, which is primarily related to the weaker stratospheric disturbance amplitude. The three types of DWs differ in spatiotemporal evolutions of the NAM and NAO pattern, different forcing by planetary waves, and varying ratios between displacement and split. This study reveals the diversity of the DWs and distinguishes their potential impacts on both continents in the Northern Hemisphere.



## 1 Introduction

In the northern winter stratosphere, the polar temperature increases dramatically in just a few days when

the polar vortex deforms or even collapses, and meanwhile the circumpolar westerly winds decrease

suddenly and even reverse the direction (Baldwin et al., 2021). This phenomenon, known as sudden

stratospheric warming (SSW), is an important manifestation of stratosphere-troposphere coupling in

spring and winter. The SSW occurrence is associated with strong, upward-propagating planetary waves

from the troposphere (Sjoberg and Birner, 2012; Butler et al., 2015). Recent studies also found that

stratospheric preconditioning might play a decisive role in inducing the SSW event, by determining the

intensity of the interaction between planetary waves and the mean flow (White et al., 2019; Yang et al.,

2023). Polvani and Waugh (2004) suggested that SSWs are caused by a sharp increase in the amplitude

of planetary-scale waves (primarily wave 1 and 2) propagating upward from the troposphere and

perturbing the stratospheric polar vortex. Modeling evidence shows that the wave 1 forces the

displacement of the polar vortex from the North Pole (Lindgren and Sheshadri, 2020), while the wave 2

determines the extent of vortex elongation and deformation (Baldwin et al., 2021).

Stratospheric variability associated with SSWs affects the near surface weather and climate through

stratosphere-troposphere coupling processes (Hitchcock and Simpson, 2014; Wu and Reichler, 2019).

Several mechanisms have been proposed to explain the possible downward impact of the stratosphere on

the troposphere, including wave-flow interaction theory (Kuroda and Kodera, 1999), balanced flow

dynamics theory (Black, 2002; Haynes, 1991), planetary wave refraction theory (Schmitz and Grieger,

1980), non-local potential vorticity response (Ambaum and Hoskins, 2002), and isentropic atmospheric

meridional mass circulation theory (Cai and Shin, 2014). The wave-mean flow interaction theory

suggests that upward-propagating tropospheric wave forcing fluctuations affect stratospheric mean flows,

which in turn affect the vertical propagation of planetary waves (Kuroda and Kodera, 1999; Hartmann,

2000). As a consequence, the stratospheric disturbances cause significant downward impact on

tropospheric circulation variations and near surface climate anomalies (Colucci and Kelleher, 2015;

Dall'Amico et al., 2010), which are proposed to be amplified by tropospheric eddy feedback (Kidston et

al., 2015).

SSWs can have a considerable impact on the troposphere and a sustained effect on surface weather for

weeks or even months (Domeisen et al., 2020; Rao et al., 2021; Lu et al., 2023). The weakened polar

vortex during SSW usually projects onto the negative phase of the Northern Annular Mode (NAM) and/or



the North Atlantic Oscillation (NAO), which gradually propagates downward (Karpechko et al., 2017; Kunz and Greatbatch, 2013; White et al., 2019). The negative phase of the NAM is often accompanied

by equatorward shifts in storm paths and tropospheric jets (Kidston et al., 2015), shifts in the centre of the East Asian jets, variation in the blocking frequency (Anstey et al., 2013), increased possibility of cold air outbreaks over Eurasia and North America (Baldwin and Dunkerton, 2001; Lehtonen and Karpechko, 2016; Lu et al., 2022; Yan et al., 2022), and likelihood of extreme rainfall (Karpechko and Manzini, 2012). The common characteristics of SSWs have been widely reported in literature (see Baldwin et al., 2021

and references therein). However, every SSW has its individual features and displays strong particularity (Karpekho et al., 2018; Rao et al., 2018, 2019, 2021; Lu et al., 2023, 2024). It is the differences between individual SSWs and the background conditions in observations that distinguish their influence on the troposphere. As a consequence, the tropospheric response signals vary in extent, area and scope. Possible factors explaining the SSW individuality especially in its influence include the SSW strength, the initial

warming location or the warming center (Zhang and Chen, 2019; Yan et al., 2022), the warming duration time (Hitchcock and Simpson, 2014), details of the wave flux between the troposphere and stratosphere (Shi et al., 2024), and the geometry of the polar vortex (split or displacement) (Maycock and Hitchcock, 2015; Rao et al., 2020). Therefore, the SSWs can be classified into different types based on those metrics. Different SSW classifications are widely used in literature. For example, based on vortex geometry shape

the SSWs are classified as split, displacement, and mixed types (Charlton and Polvani, 2007; Rao et al., 2019). Based on whether the stratosphere reflects planetary waves during the westerly recovery phase following the SSW onset, it can be divided into absorbing and reflecting types (Kodera et al., 2016). According to whether the event has had a significant impact on the troposphere, it can be grouped into downward (DW) or non-downward (NDW) types (Jucker, 2016; Runde et al., 2016; Karpechko et al.,

2017; White et al., 2019; Chwat et al., 2022). Although the DW SSWs show dripping NAM signals from the troposphere to the stratosphere, the near surface response structure is still different among DWs, with cold extreme sometimes only appearing in Eurasia, sometimes only in North America, and sometimes in both continents. However, there is still not a widely accepted subclassification for DWs events based on the coverage of the near-surface cold anomalies associated with SSW. This study is mainly concerned

with two questions: (1) What causes the inter-case difference in the DW influence on the troposphere although by definition the NAM during DWs shows downward propagation to the lower troposphere? (2) What can we learn from the flavor identification for DWs?

The organization of the paper is constructed as follows. Following the introduction, Section 2 describes the data and methods used in this paper. Section 3 compares the tropospheric response characteristics of

various DW events. Section 4 analyzes the dynamics related to various DW events. Finally, Section 5 provides a summary and discussion.

## 2 Data and methods

### 2.1 Reanalysis data

We use the European Centre for Medium-Range Weather Forecasts (ECMWF) fifth generation reanalysis

(ERA5) dataset over the period from October 1940 to December 2022. The data prior to 1958 are used with an expectation of selecting more SSW samples. Both three-dimensional and two-dimensional data are used. The three-dimensional variables on the isobaric levels include the air temperature, the geopotential, the zonal and meridional winds, and the vertical velocity on p coordinate. The two-dimensional variables at a single surface level include total precipitation and the temperature at two

meters (t2m). The isobaric levels extend from 1000 hPa to 1 hPa, and the local horizontal resolution of the ERA5 is 0.25° latitude by 0.25° longitude (this dataset is collected at 1.5° latitude by 1.5° longitude for easy handling).

Considering that the SSW samples from the ERA5 reanalysis are not too large, three historical simulations (r1i1p1f1, r2i1p1f1, and r3i1p1f1) from the CESM2-WACCM are employed to increase the

SSW number. The CESM2-WACCM model has been shown to well simulate the SSWs and the downward impact (e.g., Liang et al., 2022). The historical simulation starts from 1850 and ends in 2014, which is one of the common experiments from CMIP6. Outputs from CESM2-WACCM have a horizontal resolution of 0.9° latitude by 1.25° longitude at 19 standard pressure levels.

The daily climatology is computed as the long-term mean for each calendar day, and the raw daily

climatology is smoothed using 31-day means. The daily anomalies refer to the detrended deviation relative to the smoothed daily climatology.

### 2.2 Methods

#### 2.2.1 DW and NDW event definitions

We use the definition from Charlton and Polvani (2007) to identify SSWs. Specifically, all days when a

change in the zonal-mean zonal wind $\bar{u}$ from westerlies to easterlies occurs at 60°N and 10 hPa within



the period from 1 November to 31 March each year are selected. The first day when $\bar{u}$ undergoes a transition from westerlies to easterlies is defined as the onset date of this SSW. Several SSW definitions considering the wind reversal at different latitudes and heights in the circumpolar region have been compared by Butler and Gerber (2018). They concluded that a small modification in the wind position

or the wind threshold used for the SSW definition does not lead to major changes in the statistics of the SSWs. Considering the zonal winds change radically even after the SSW onset, the onset date of the two SSW events must be more than 20 days of consecutive westerlies apart (roughly double the radiative timescales in the middle stratosphere). Further, $\bar{u}$ must have recovered to westerlies at least for 10 consecutive days prior to 30 April, which can exclude final warmings (e.g., Charlton and Polvani, 2007).

Finally, 52 SSWs are selected from ERA5, and 273 SSWs are selected from CESM2-WACCM. The average occurrence frequency of SSW is approximately 0.63 per year in ERA5 and 0.55 per year in CESM2-WACCM, consistent with previous studies (Liu et al., 2019; Rao et al., 2021).

To classify the SSW as the DW or the NDW, the NAM index is calculated using the reanalysis (Baldwin and Thompson, 2009). After removing the daily climatological mean from the geopotential heights, the

anomaly data are area-weighted (i.e., multiplied by cosine of the latitude) over 20°-90°N. The anomaly data are normalized for each pressure level, and the empirical orthogonal function is performed to extract the leading mode (i.e., the NAM pattern). The anomaly field is projected onto the leading mode to compute the corresponding time series (i.e., the NAM index). The NAM index for each pressure level is calculated separately, and the NAM at 150 and 850 hPa are used to identify DWs.

The definition of DWs used in this paper follows the method by Natarajan et al. (2019). Namely, a DW is defined when the SSW satisfies the following three conditions: During the 45-day period from day 8 to day 52 relative to the SSW onset date, 1) the average NAM index is negative at both 1000 hPa and 150 hPa; 2) the percentage of days with a negative NAM index is greater than 70% at 150 hPa; and 3) the percentage of days with a negative NAM index is greater than 50% at 1000 hPa. To reduce the effect

of topographic complexity over high-terrain regions, White et al. (2019) adjusted 1000 hPa to 850 hPa for the first and third conditions, which are also adjusted in this study. Based on these criteria, 33 DWs (63% of all SSWs) are identified in ERA5, compared to 119 DWs (44% of all SSWs) from CESM2-WACCM.



### 2.2.2 Classification of DWs

It is assumed that following the DW event onset, continental cold anomalies can develop over Eurasia

and/or North America, implying an increase in cold air outbreaks after the DW SSW onset. To better

describe and distinguish the DWs, we further divide DWs based on the inland temperature anomalies

within 40 days after the onset of DWs. Considering that cold surges are more active over the Eurasia

(Europe + Asia) and North America (US + Canada) following the DWs, a comparison between the cold

anomalies over the two regions can further classify the DWs into three types: cold anomalies only

appearing over North America (NA), only appearing over Eurasia (EA), and over both regions (BOTH).

Regions focused in this study include Europe (40°-70°N, 0°-60°E), Asia (40°-70°N, 60°-140°E), United

States (30°-46°N, 70°-120°W), and Canada (46°-60°N, 60°-120°W). Within one region, the area-

averaged temperature anomalies over a 40-day period are considered to be associated with the downward

impact of the DW if the anomalies meet the following two criteria: 1) the percentage of days with cold

anomalies is greater than 50%; 2) the mean temperature anomaly over a 40-day period is less than -0.3°C.

The classification of SSWs into split and displacement events is based on the method initially developed

by Mitchell et al. (2011) and adapted by Seviour et al. (2013) using the two-dimensional (2D) moment

analysis method. Maycock and Hitchcock (2015) modified this analysis method by using the geopotential

height at 10 hPa instead of the potential vorticity used in Mitchell et al. (2011). Several modifications for

some parameters in this study are as follows. The first is the edge of the polar vortex, which we define

as the climatological mean geopotential height over 60°N and 10 hPa in winter (e.g., Maycock and

Hitchcock, 2015). The second is the thresholds for the split and displacement SSWs. We choose the

thresholds as the most equatorward 5.7% of centroid latitudes and largest 5.2% of aspect ratios, yielding

a threshold of 62.9°N for centroid latitude and 2.46 for aspect ratio, respectively (e.g., Mitchell et al.,

2011; Seviour et al., 2013). Table 1 shows the onset date and specific categorization of each event in

ERA5. The general statistics of SSWs from the three historical simulation members by CESM2-WACCM

are shown in Table 2.

**Table 1**. Statistics of the SSWs from 1940–2022. The second column (S/D) shows the SSW type based
on the vortex shape (D=displacement, S=split). The third column shows the SSW type based on whether
the stratospheric signal propagates downward (DW=downward, NDW=non-downward). The last column
shows the subclassification of DWs.

| onset date | S/D | DW/NDW | type | onset date | S/D | DW/NDW | type |
|---|---|---|---|---|---|---|---|
| 1941-02-07 | D | DW | BOTH | 1981-03-04 | D | NDW | — |



| 1942-03-21 | D | NDW | — | 1981-12-04 | D | NDW | — |
|---|---|---|---|---|---|---|---|
| 1945-03-19 | S | NDW | — | 1984-02-24 | D | DW | BOTH |
| 1946-02-19 | D | NDW | — | 1985-01-01 | S | DW | BOTH |
| 1946-03-19 | D | NDW | — | 1987-01-23 | D | DW | EA |
| 1950-03-05 | D | NDW | — | 1987-12-08 | S | NDW | — |
| 1952-02-22 | S | DW | BOTH | 1988-03-14 | S | NDW | — |
| 1952-11-19 | D | DW | EA | 1989-02-21 | S | NDW | — |
| 1954-12-18 | D | DW | EA | 1998-12-15 | D | NDW | — |
| 1955-01-26 | S | DW | BOTH | 1999-02-26 | S | DW | EA |
| 1957-02-04 | S | DW | EA | 2000-03-20 | D | NDW | — |
| 1958-02-01 | D | DW | NA | 2001-02-11 | S | DW | NA |
| 1960-01-17 | D | DW | BOTH | 2001-12-30 | D | NDW | — |
| 1963-01-27 | S | NDW | — | 2002-02-17 | D | DW | NA |
| 1965-12-16 | D | NDW | — | 2003-01-18 | S | NDW | — |
| 1966-02-22 | S | DW | EA | 2004-01-05 | D | DW | NA |
| 1968-01-07 | S | DW | EA | 2006-01-21 | D | DW | EA |
| 1968-11-28 | D | DW | BOTH | 2007-02-24 | D | NDW | — |
| 1969-03-13 | D | NDW | — | 2008-02-22 | D | DW | NA |
| 1970-01-02 | D | DW | BOTH | 2009-01-24 | S | DW | EA |
| 1971-01-18 | S | DW | BOTH | 2010-02-09 | S | DW | EA |
| 1971-03-20 | D | DW | BOTH | 2010-03-24 | D | DW | EA |
| 1973-01-31 | S | NDW | — | 2013-01-06 | S | DW | EA |
| 1977-01-09 | S | DW | BOTH | 2018-02-12 | S | DW | EA |
| 1979-02-22 | S | DW | BOTH | 2019-01-01 | S | DW | NA |
| 1980-02-29 | D | DW | BOTH | 2021-01-05 | D | DW | EA |

Table 2. general statistics of SSWs from the three historical simulation members by CESM2-WACCM.

| Member | SSW | DW | | | NDW |
|---|---|---|---|---|---|
| | | BOTH | EA | NA | |
| **r1i1p1f1** | 90 | 13 | 18 | 9 | 50 |
| **r2i1p1f1** | 87 | 11 | 15 | 10 | 51 |
| **r3i1p1f1** | 96 | 16 | 18 | 9 | 53 |
| **Total** | 273 | 40 | 51 | 28 | 154 |


### 2.2.3 Isentropic potential vorticity

In this paper, isentropic potential vorticity (IPV) is used. The vertical component of IPV is defined

(Hoskins et al., 1985) as:

$$\text{IPV} = -\frac{g\left(f + \vec{k} \cdot \nabla_\theta \times \vec{V}\right)}{\frac{\partial p}{\partial \theta}}, \tag{1}$$

where $f$ is the Coriolis parameter, $\theta$ is the potential temperature, $g$ is gravity acceleration, $p$ is air



pressure, $\vec{k}$ is the vertical unit vector, and $\vec{V}$ is horizontal wind vector at isentropic surface.

**2.2.4 Eliassen-Palm (E-P) flux**

The E-P flux and its divergence can characterize the propagation of quasi-geostrophic planetary waves and its interaction with the mean flows. The E-P flux and its divergence are employed to diagnose the

dynamical processes during SSWs as follows (Edmon et al., 1980):

$$F_\varphi = -a(cos\,\varphi)\overline{u'v'}, \tag{2}$$

$$F_p = a(cos\,\varphi)f\frac{\overline{v'\theta'}}{\overline{\theta_p}}, \tag{3}$$

$$\nabla \cdot \boldsymbol{F} = \frac{1}{a\,cos\,\varphi}\frac{\partial\left(F_\varphi\,cos\,\varphi\right)}{\partial\varphi} + \frac{\partial F_p}{\partial p}, \tag{4}$$

where $F_\varphi$ is the horizontal component of the E-P flux, $F_p$ is the vertical component, and $\nabla \cdot \boldsymbol{F}$ is the

divergence of the E-P flux, and $a$, $\varphi$ are the Earth's radius and latitude, respectively. The E-P flux vector characterizes the propagation direction of the planetary waves. The E-P flux divergence indicates the effect of the planetary waves on the mean flow. When the E-P flux is convergent (divergent), it indicates that there is easterly (westerly) forcing of the mean flow.

**3 Comparison of tropospheric responses to three types of DWs**

**3.1 Spatiotemporal evolution of the NAM**

The NAM index can be used to describe the downward propagation of stratospheric disturbance. The composite evolutions of the NAM index for the three types of DWs are compared in Fig. 1. Consistent with previous study (Karpechko et al., 2017; Kunz and Greatbatch, 2013; White et al., 2019), negative NAM signals appear above 200 hPa within a few days around the SSW onset date. After the DW onset,

the negative signal propagates downward into the troposphere (Fig. 1a1-c1), forming a typical dripping-paint pattern in the troposphere (e.g., Baldwin and Dunkerton, 2001). On day 20 and afterward, the upper stratosphere gradually recovers to the positive NAM, while the negative signal below 50 hPa returns to the positive NAM at different times for the three types of DWs. The negative NAM signal below 50 hPa can persist until day 60 and even afterward, suggesting that the downward influence of DWs is persistent,

providing a possible predictability source for the troposphere (Rao et al., 2021; Lu et al., 2023). Differences in the NAM evolutions are noticed for the three types of DWs in ERA5.

1) In the pre-SSW onset period, the NAM displays different behaviors. The positive NAM mainly



develops around day -50 and day -30 with the most significant signals in the lower stratosphere and upper troposphere for the type BOTH. The positive NAM only develops in the upper stratosphere around day -30 for the type EA. In contrast, the positive NAM dramatically develops from day -50 in the upper stratosphere to day -15 in the lower troposphere, displaying a noticeable downward propagation for the type NA.

2) In the post-SSW onset period, the negative NAM is structured in different spatiotemporal dripping shapes. Around 20 days or more after the SSW onset, the NAM is reversed in the upper stratosphere, while the reversion time in lower levels is later. The negative NAM at 50 hPa persists longer than any other level for all types of DWs and even NDW. In contrast, the NAM sign reversion is earliest for NA out of the three DW types around day 50 at 50 hPa (Fig. 1c1), while the NAM sign change is much slower for BOTH and EA beyond day 60 (Fig. 1a1, b1). It is also seen that the NAM sign change for NDW is also around day 50 at 50 hPa (Fig. 1d1).

3) The positive NAM intensity before the SSW onset and the negative NAM intensity after the SSW onset are variously contrasted for the DWs and NDW. The NAM evolves from moderately positive to moderately negative for the type BOTH. It evolves from weakly positive to weakly but persistently negative for type EA. Further, it evolves from intensely positive to intensely but shortly negative for type NA. The positive NAM develops 40 days or more before the NDW, but the negative NAM intensity and the NAM contrast before and after the NDW onset are weak.

4) The near surface exhibits different behaviors in the NAM intermittent signals for the three types of DWs. Specifically, the negative NAM at the near surface is continuous for BOTH and EA types, while it is very short in the persistent time and shows a low significance level for NA. At the very beginning of the NA DWs, the NAM is still positive at the near surface due to the lagged downward impact of the stratosphere, with the negative NAM from day 10 to day 50. In contrast, the negative NAM signal fails to appear at the near surface for NDWs, only with significant positive NAM around day 30. The difference between BOTH and NDW and between NA and EA is most significant in the troposphere and near surface (Fig. 1e1, f1), implying a diversity in the persistency and NAM intensity among SSW types. Comparing CESM2-WACCM simulations (Fig. 1a2-d2) with ERA5, the evolution of the NAM is highly consistent. The conclusions are nearly unchanged if the reanalysis is replaced by model simulations: the downward propagation of the NAM is more continuous during EA than NA, and the NAM is much shallower during the NDW than during DWs.

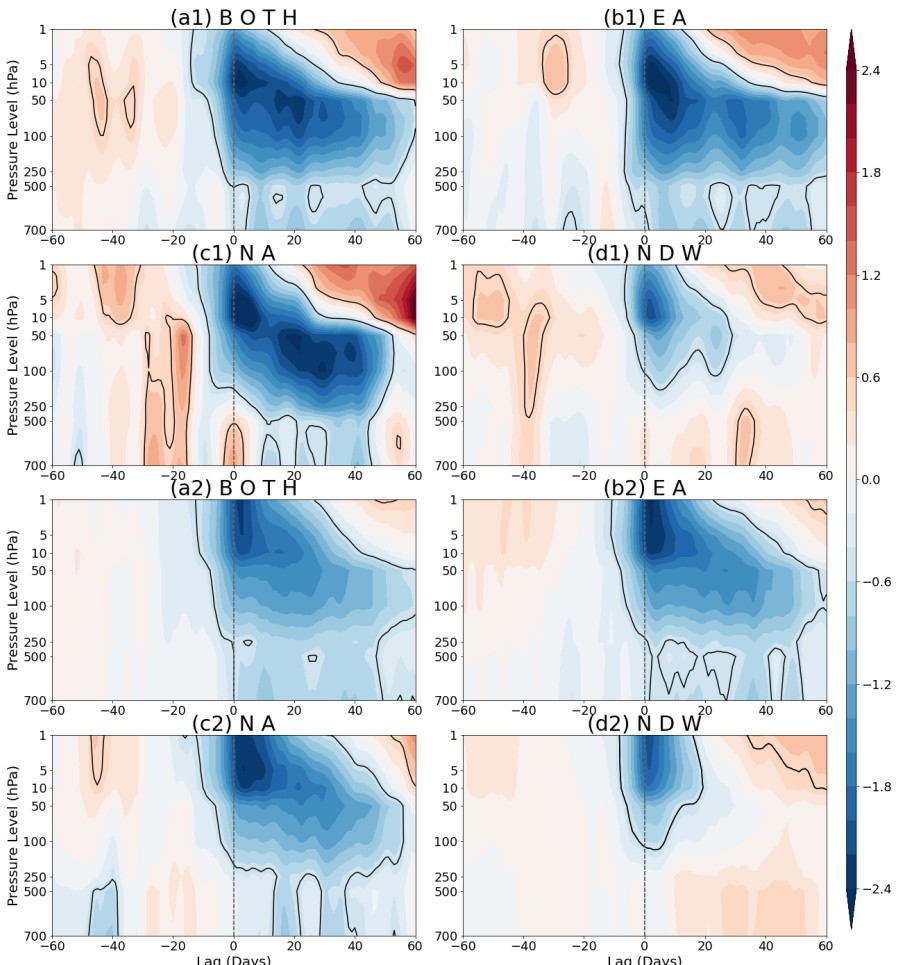

**Figure 1.** The composite evolution of the NAM index for (a-c) three types of DWs and (d) NDW events.

(a) BOTH, (b) EA, (c) NA, (d) NDW. The reanalysis is numbered with "1" following the plot letters, and the model simulation is numbered with "2". The units are in standard deviations. The black line represents statistical significance at the 95% level for the composite.

The NAO index is closely related to low-level circulation over the North Atlantic, which is significantly correlated with the probability of extremes, such as regional coldness, snowstorm, and strong winds (Thompson and Wallace, 2001; Scaife et al., 2014). Fig. 2 compares the probability density functions (PDFs) of the 60-day average NAO index after the onset for each type of event. Comparisons with a random sample of 2000 winters show that the probability of the NAO index for DWs shifts left toward



negative, which indicates that DWs have a significant effect on the surface circulation by modifying the

PDF of the NAO. In ERA5 (Fig. 2a), the NAO index after the DW onset for BOTH is 2-3 times more

likely to be less than -0.8 than for other cases, with the median roughly located at -0.7, and the probability

that the index is positive is almost zero. The median values of the NAO after EA and NA are the same,

roughly located at -0.5, but the difference is noticeable. Namely, the PDF of the NAO for NA is more

dispersed and the PDF peak is smaller than that for EA and BOTH. In contrast, the type BOTH showed

the largest composite mean of NAO (-0.762), followed by EA (mean NAO = -0.567) and NA (mean

NAO=-0.435). The PDF of NAO for NDW is primarily concentrated between -0.2 and 0.4 (68.4%), with

median and mean values near 0.1 and 0.088, which might indicate that the NAO nearly has no preference

during NDW events.

In the model simulation (Fig. 2b), the probability density function distribution of the NAO indices also

highly resembles that in ERA5 (Fig. 2a). Namely, the mean probability density function for EA is much

more left-skewed than that for BOTH and NA. Consistent with ERA5, the probability density function

for NA is distributed more flattened than that for BOTH and EA. The probability density function

distribution is almost symmetric on both sides of zero for NDW in both ERA5 and CESM2-WACCM,

although the peak value (~1.1) is larger than that for the climatological probability density function (~0.8).



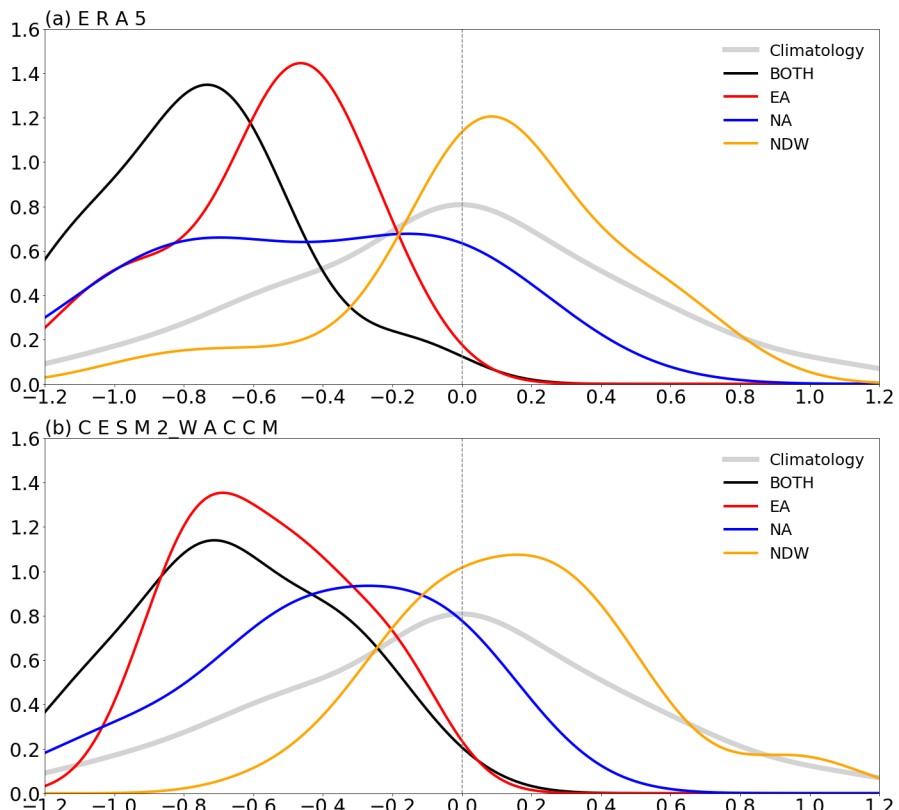

**Figure 2.** Probability density functions for the average NAO index during 2000 randomly averaged 60-day periods in winter (grey curve), the 60 days after SSW onset date for BOTH (black curve), EA (red curve), NA (blue curve), and NDW (orange curve) events from two datasets: (a) ERA5 and (b) historical simulations by CESM2-WACCM. The Kernel Density Estimation is used for the 60-day mean index to smooth the PDF curve.

### 3.2 Comparison of near surface response

To compare the possible different impact of DWs on the near surface, the composite t2m anomalies in the 40-day intervals before and after the SSW onset are shown in Fig. 3. In both ERA5 and CESM2-WACCM simulations (Fig. 3a-d), the cold anomalies appear over different regions during the SSW for DWs and the NDW. Specifically, in the pre-SSW period (day -40 to day 0), significant cold anomalies have well developed over northern Eurasia for BOTH and EA, while the cold anomalies are not detectable over Eurasia for NA (Fig. 3a-c). Although cold anomalies also appear over Eurasia, the



anomaly magnitude is fairly weak for NDWs (Fig. 3d). The western coast of North America is covered with cold anomalies for BOTH and NDW (Fig. 3a, d), while significant warm anomalies appear over

most of North America for EA and NA (Fig. 3b, c). It is also found that significant warm anomalies develop over Europe for NA (Fig. 3c). For BOTH and EA, an anomalous high appears over the Arctic at 500 hPa, while for NA two anomalous high centers appear over North America and Europe, respectively (Fig. 3a-c). The tropospheric circulation anomalies are much weaker for NDWs than DWs (Fig. 3d). The general patterns of t2m and 500-hPa height anomalies are highly consistent between ERA5 and CESM2-

WACCM simulations.

In the post-SSW period, the t2m anomaly pattern is contrastingly different for the three types of DWs and the NDW, consistent with the construction for t2m anomalies in ERA5 and CESM2-WACCM. For the type BOTH, the cold anomalies in the Eastern Hemisphere continue to strengthen and expand southward, covering most of Eurasia; the cold anomalies in the Western Hemisphere expand from

western coasts of North America to most of the continent (Fig. 3e). The positive height anomaly center for the type BOTH is located over Greenland and the Bering Strait in ERA5 and over Greenland in model simulations. For the type EA, the cold anomalies persist over northern Eurasia and the cold center moves further eastward to Northeast Asia, while the warm anomalies over eastern Canada and southern Europe strengthen (Fig. 3f). The positive height anomaly center remains over the Arctic Ocean and Canada. For

the type NA, warm anomalies over eastern Europe also intensify, while the warm anomalies over North America are replaced by anomalous coldness with the center over the central US (Fig. 3g). The positive height anomaly centers move from Canada and Europe to Greenland and Central Asia, respectively, although the composite anomalies are relatively weaker in CESM2-WACCM. For the NDW, the t2m and 500-hPa height anomalies over both lands are very scattered and less organized, although patches of

warm anomalies are observed over Asia (Fig. 3h).



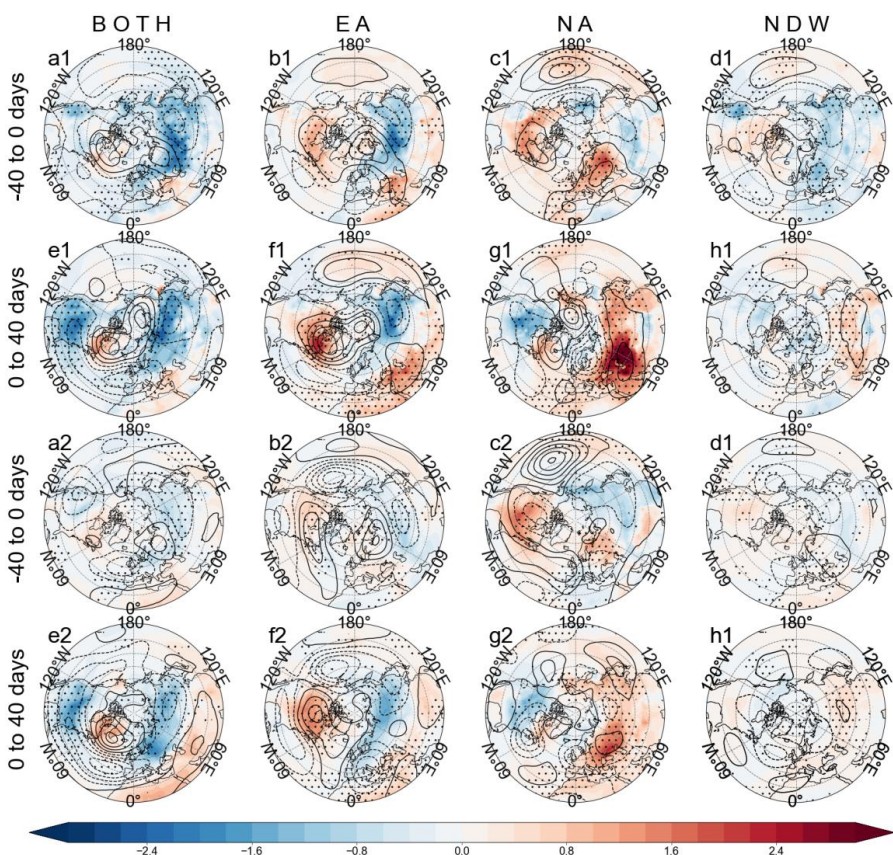

**Figure 3.** Composite 2-meter temperature (t2m) anomalies (shadings; units: K) and 500-hPa geopotential height anomalies (contours; units: gpm) for (a, e) BOTH, (b, f) EA, (c, g) NA and (d, h) NDW events before the SSW onset (top row) and afterward (bottom row) from two datasets: (a1-h1) ERA5 and (a2-h2) historical simulations by CESM2-WACCM. The composite is based on the mean of 40-day intervals. The dots mark the composite anomalies at the 95% confidence level using the *t*-test.

To well depict the dynamic process of the near surface response to different types of SSWs, the composite evolution of t2m anomalies over Eurasia and North America is shown in Fig. 4. In both datasets, for the type BOTH, cold anomalies develop throughout the SSW occurrence period from day -40 to day 40 over all the four regions (Europe, Asia, US, and Canada), except that the negative t2m anomalies at Canada are relatively weak and exhibit relatively large subseasonal variability (Fig. 4a). For the type EA, cold anomalies are persistent over northern Eurasia, larger in Asia than in Europe, while warm anomalies are



persistent over North America, larger in Canada than in US (Fig. 4b). For the type NA, the reversal of

temperature anomaly sign is observed over North America (US and Canada). The reversal of temperature

anomalies from positive to negative indicates outbreaks of cold air (Lehtonen and Karpechko, 2016).

Namely, cold air outbreaks increase in North America, while the anomalously cold state gradually

recovers to normal in Asia with moderately warm state in Europe (Fig. 4c). For NDWs, the temperature

anomalies are fairly weak throughout the SSW onset except that Canada experiences a transition from

anomalously warm state to cold (Fig. 4d).

Comparing ERA5 and historical simulations by CESM2-WACCM, the composite t2m anomaly

amplitude in model simulations is relatively weak and less significant in the pre-SSW period. Consistent

with ERA5, cold anomalies persist from pre-SSW to post-SSW periods over both continents for the

BOTH type in the model simulations. Further, the persistent cold anomalies over Eurasia from pre-SSW

to post-SSW periods are also present in model evidence for the EA type. The reversal of t2m anomalies

from positive to negative over North America and from negative to positive over Eurasia is also

confirmed by model simulations for the NA type. Both ERA5 and CESM2-WACCM simulations show

that the t2m anomalies over both continents are weak and insignificant most of the time during the SSW.



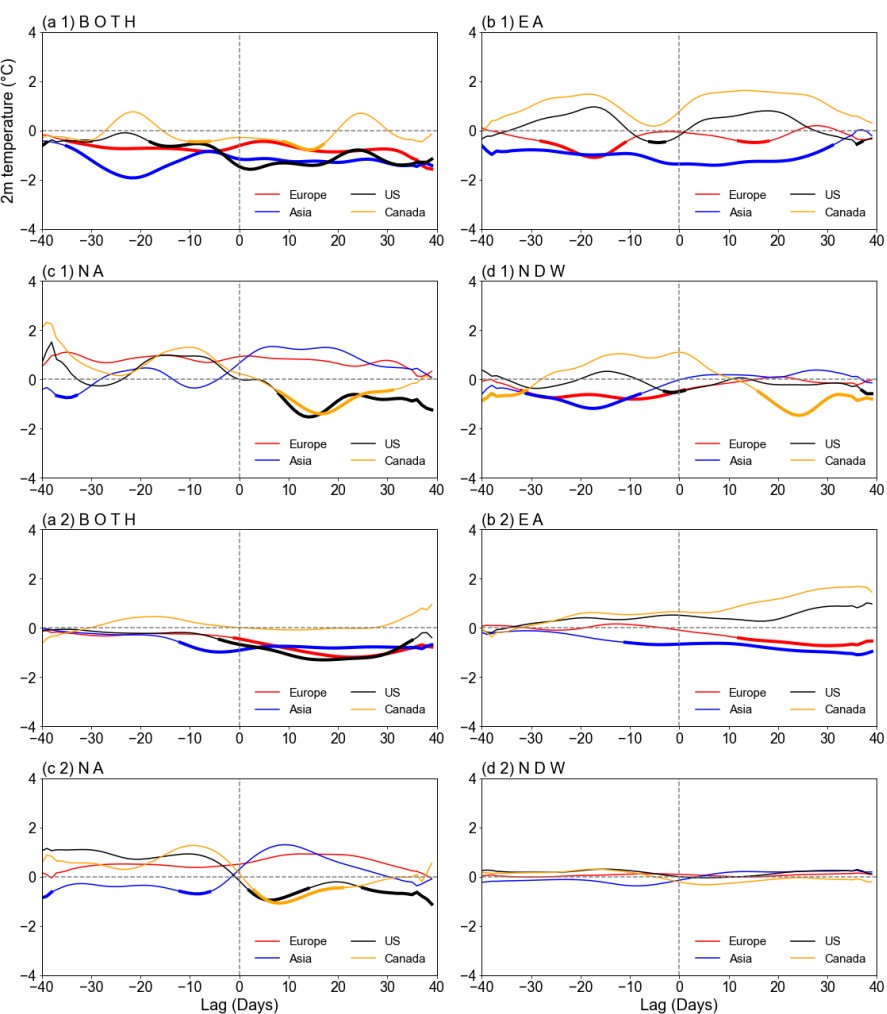

**Figure 4.** Evolution of the area-averaged t2m anomalies (units: K) from day -40 to day 40 over four regions (I–IV), including Europe (40°-70°N, 0°-60°E), Asia (40°-70°N, 60°-140°E), United States (30°-46°N, 70°-120°W), and Canada (46°-60°N, 60°-120°W) for (a) BOTH; (b) EA; (c) NA and (d) NDW events from two datasets: (a1-d1) ERA5 and (a2-d2) historical simulations by CESM2-WACCM. The thickened part of the line denotes the composite at the 95% confidence level.

### 3.3 Analysis of isentropic potential vorticity

In order to further explore the process of cold air activity, an analysis of the isentropic potential vorticity (IPV) is investigated for the three types of DWs and the NDW. According to Eq. (1), the IPV is



proportional to the absolute vorticity and static stability. In the case of equal vorticity, the cold air mass usually has a higher potential vorticity value due to its relatively larger static stability, which can be used

to track the cold air activity on the isentropic surface (Hoskins et al., 1985; Lu and Ding, 2015). The 315K isentropic level has been used to track the cold air sources by detecting the movement of high IPV center (Jeong et al., 2006). The composite evolutions of IPV anomalies are shown in Fig. 5 for all types of events based on ERA5 and CESM2-WACCM simulations. For the type BOTH, a patch of high IPVs appear from North Pacific 30-20 days prior to the event and gradually move southeastwards in the

following period, which finally reach the North America for both datasets (Fig. 5a). During day 10 to 20, anomalously high IPV air moves to 45°N and even more southward, with the anomaly amplitude gradually weakened. Significant negative IPV anomalies develop over Greenland, consistent with the local anomalous high (see Fig. 3e), which helps advect cold air to eastern North America. In contrast, there are two high IPV centers in Eurasia, one over North Atlantic, and the other over Central Asia. The

anomalously high IPV center over North Atlantic gradually moves to Europe, indicating the cold air outbreak. The high IPV center over Central Asia is more stable from day -30 to -10 and redevelops from day 0 to 20.

For the EA events, the high IPV center in the North Atlantic is weak and significant high IPV anomalies appear in Europe (Fig. 5b). The high IPV anomalies develop in North Atlantic soon after the SSW onset,

and the Arctic is nearly covered by the positive IPV anomalies. Another positive IPV anomaly center is detected over North Asia since day -30 to -20, which diminishes from day -20 to -10 and redevelops from day -10 to 10. The high IPV patch moves southeastward to East Asia after day 10, indicating cold air outbreaks in local regions.

For the NA events, the high IPV center first appears over North Pacific from day -20 to -10, which is still

active from day -10 to 10 (Fig. 5c). The positive IPV anomaly intensity weakens from day 0 to 10 and then redevelops in the later period. Further, it is also noticed that a large patch of positive IPV anomalies form over North Atlantic after the SSW onset, which is nearly motionless. Similar to BOTH events, the blocking pattern denoted as a wide range of negative IPV anomalies over Greenland is observed, and the upstream winds can advect cold air to eastern North America.

For the NDW events, the IPV anomalies are relatively weak, and the positive IPV anomalies are scattered and less organized over the land (Fig. 5d), consistent with the distribution of t2m anomalies. Weak positive IPV anomalies primarily appear over the oceans and are nearly motionless. It is also observed





that a narrow band of positive IPV anomalies exists from the west to east across Canada during day 20 to day 30.

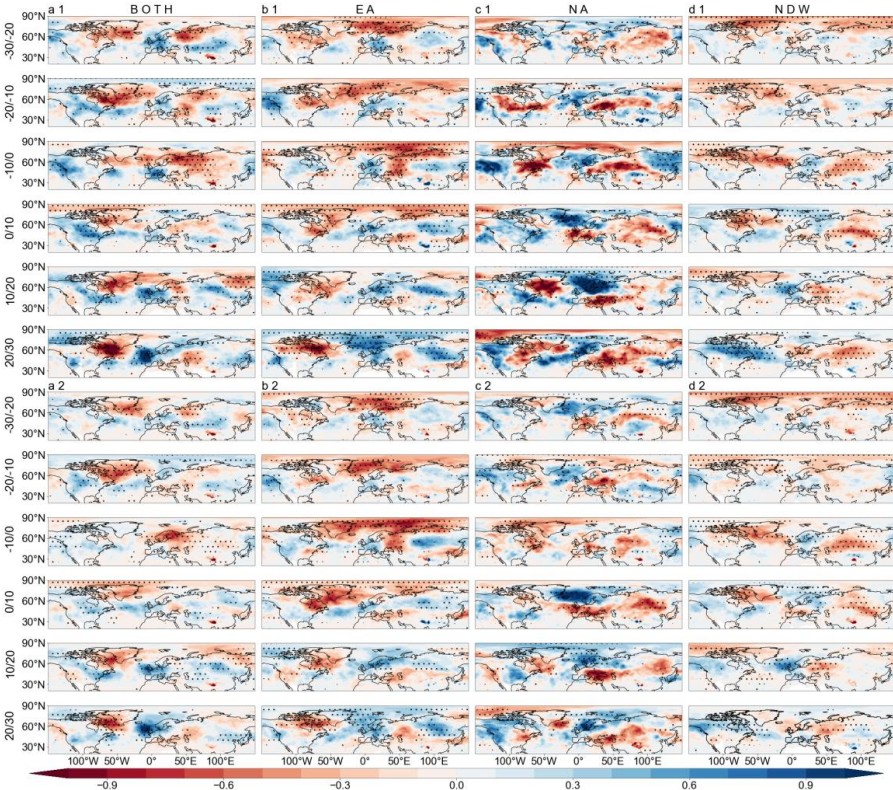


**Figure 5.** Composite isentropic potential vorticity (IPV) anomalies (shadings; units: PVU, 1 PVU = $10^{-6}$ m$^2$ K kg$^{-1}$) at 315 K for (a) BOTH, (b) EA, (c) NA, and (d) NDW events from two datasets: (a1-d1) ERA5 and (a2-d2) historical simulations by CESM2-WACCM. The composite is based on the mean of 10-day intervals. The dots mark the composite anomalies at the 95% confidence level using the *t*-test.

**3.4 Total precipitation**

Previous studies have shown that SSWs not only influence the cold air outbreaks, but also affect the rainfall anomalies (King et al., 2019; Oehrlein et al., 2021). Negative NAM phases are usually accompanied by a possible equatorward shift of the storm track, and a possible intensification of the moving cyclone at low latitudes (Afargan-Gerstman and Domeisen, 2020; McAfee and Russell, 2008;

Thompson and Wallace, 2001). The composite precipitation anomalies from day 0 to 40 are shown in Fig. 6 to compare the rainfall anomaly sensitivity to the SSW type. The most pronounced common



features for the four conditions in both ERA5 and CESM2-WACCM simulations are the rainfall dipole

structure over North Atlantic - Europe. Positive rainfall anomalies form over Atlantic midlatitudes and

the Mediterranean Sea, while negative rainfall anomalies form over Atlantic high-latitudes. This rainfall

dipole is strong for BOTH and EA (Fig. 6a, b), while the intensity is relatively weak for NA and NDW

(Fig. 6c, d). The positive rainfall anomaly in midlatitudes for NA is broken into two chains, one over the

ocean, and the other biased toward the Eurasian land (Fig. 6c). Although the NAM fails to descend to the

near surface, the rainfall dipole is biased farther northward for NDW (Fig. 6d).

Comparing ERA5 and CESM2-WACCM simulations, the rainfall dipole is consistently present for the

BOTH and EA types (Fig. 6a, b). The negative rainfall anomaly lobe in high latitudes is much weaker

for the NA type than for the BOTH and EA types for both datasets, although the positive rainfall anomaly

band in midlatitudes is still present (Fig. 6c). In ERA5, the dipole rainfall anomalies are nearly absent

due to the non-downward propagation of NDW SSWs, while the rainfall dipole disappears in model

simulations for the NDW events (Fig. 6d).

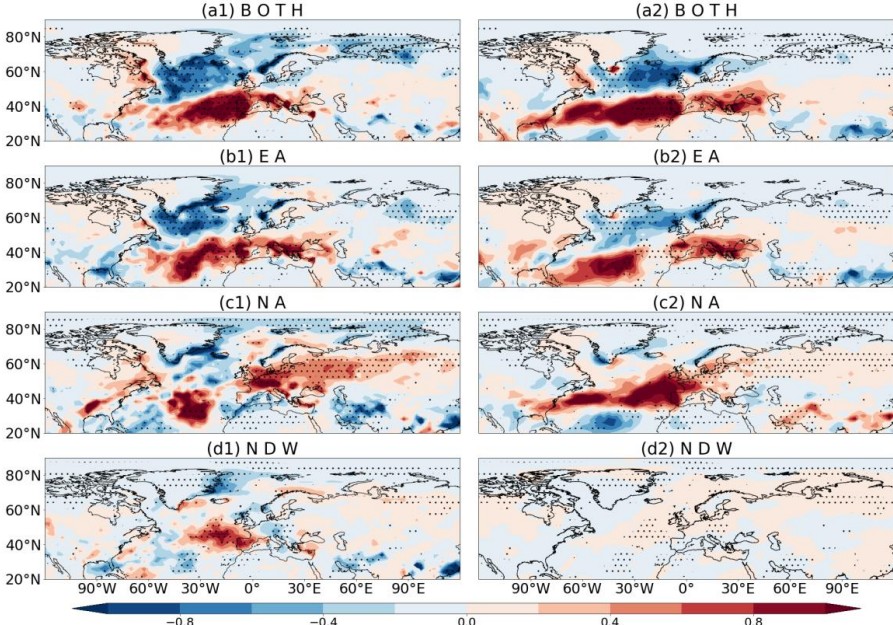


**Figure 6.** Composite precipitation anomalies (shadings; units: mm day$^{-1}$) for (a) BOTH, (b) EA, (c) NA
and (d) NDW events after the SSW onset from two datasets: (a1-d1) ERA5 and (a2-d2) historical
simulations by CESM2-WACCM. The composite is based on the mean of day 0 to 40. The dots mark the
composite anomalies at the 95% confidence level using the *t*-test.





## 4 Dynamic diagnostics

In order to compare the large-scale atmospheric dynamics between different types of SSW events and to better understand their downward influences, the composite circulation anomalies at 100 hPa and the sea level pressure anomalies are shown in Fig. 7. In the pre-SSW period, the circulation structure is differently organized for three types of DWs and the NDW based on ERA5 and historical simulations by CESM2-WACCM. For the type BOTH, an anomalous high appears over Canada, and a low anomaly center forms over central northern Eurasia (Fig. 7a), implying the displacement of the polar vortex toward North Asia. On the near surface, the negative NAM pattern has well developed, positive MSLP anomalies prevail over the Arctic, and negative anomalies develop over midlatitudes. The height anomaly distribution at 100 hPa for the type EA is similar to that for BOTH except that the height anomaly centers are further eastward situated (Fig. 7b). The anomalous high is situated over Europe, while the anomalous low is situated over Northeastern Asia. Positive MSLP anomalies develop over the Arctic and northern Europe, while the negative MSLP anomalies form over northern Canada. For the type NA (Fig. 7c), the anomalous high at 100 hPa is centered over the Great Lakes, while the low covers most of northeast Asia. However, the near surface is covered by the anomalous low over most of the Arctic. For NDWs, the polar vortex at 100 hPa is not significantly disturbed, although the near surface exhibits a pressure anomaly dipole, with the low over the Bering Strait – Alaska and the high over the northern Europe (Fig. 7d). Comparing the three types of DWs, the precursor circulation anomalies for BOTH are more baroclinic from the lower stratosphere to the troposphere (Fig. 7a, b), while for NA and EA, the circulation anomalies are nearly barotropic in high latitudes (Fig. 7b, c). The height anomalies at 100 hPa are much weaker for NDWs (Fig. 7d) than for DWs. The circulation anomalies for the three types of DWs in the pre-SSW period are nearly barotropic in model simulations. The NAM pattern in model simulations is more clearly present in model simulations. For the type BOTH (Fig. 7a2), three anomalous low centers appear in the mid-latitudes over East Asia, western North America, and the Atlantic Ocean, respectively. For the type EA (Fig. 7b2), two anomalous high centers form in Europe and Canada, and anomalous low centers appear in the North Pacific. For the type NA (Fig. 7c2), two anomalous high centers are located over the North Pacific and North Atlantic, respectively. For NDW events (Fig. 7d2), the height anomalies are much weaker than the DWs, consistent with ERA5.

In the post-SSW period, the Arctic is completely covered by high anomalies from the near surface to the lower stratosphere with a nearly barotropic circulation structure (Fig. 7e-h). In the lower stratosphere,



the Arctic for all types of events is covered by the anomalous high, indicating the breakup of the

stratospheric polar vortex. The positive anomaly amplitude for the three types of DWs in both ERA5 and

model simulations is comparable, and their differences are mainly featured by midlatitude circulation

anomalies. For BOTH, negative height anomalies are clearly present in midlatitudes with three centers,

one over North Atlantic, one over Canada, and one over East Asia at 100hPa (Fig. 7e). Similarly, the

MSLP anomalies are well structured in a negative NAM pattern with the negative anomalies maximized

over North Atlantic along midlatitudes. For EA, the negative height anomalies are not as detectable as

for BOTH, and only the negative center over East Asia is clearly observed at 100hPa (Fig. 7f). On the

near surface, the NAM structure is more clearly present in the Atlantic sector than in the Pacific sector.

The local anomalous high over North Pacific disrupts the annular structure in the Eastern Hemisphere.

For NA, the anticyclonic anomalies at 100 hPa are further biased toward the Arctic Canada, while

nominal negative height anomaly band in midlatitudes for the negative NAM is only present over North

Pacific (Fig. 7g). Similarly, the negative MSLP anomaly band is concentrated from the Western Europe

to East Asia and North Pacific (Fig. 7g), different from the long anomalous low band in midlatitudes for

BOTH and EA. For NDWs, although positive height anomalies are present over the Arctic in the

stratosphere at 100hPa (Fig. 7h), the amplitude is nearly half or one third of the strength for DWs. The

MSLP anomalies in Arctic are very weak, while significantly negative MSPL anomalies over North

Atlantic–Northern Eurasian are observed. The circulation anomaly distribution in the post-SSW period

is consistent between ERA5 and model simulations. The negative NAM structure is more clearly present

in model simulations, possibly due to a much larger sample size in model simulations than in ERA5.

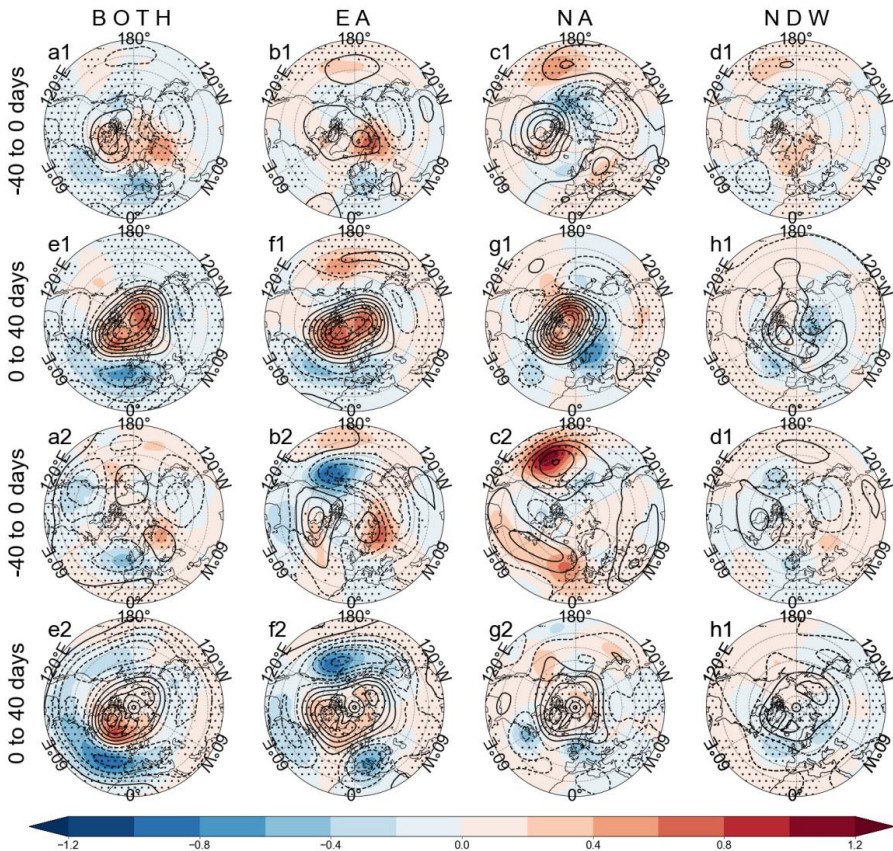


**Figure 7.** Composite 100-hPa geopotential height anomalies (contours; units: gpm) and sea level pressure

anomalies (shadings; units: hPa) for (a, e) BOTH, (b, f) EA, (c, g) NA and (d, h) NDW events before the

SSW onset (top row) and afterward (bottom row) from two datasets: (a1-h1) ERA5 and (a2-h2) historical

simulations by CESM2-WACCM. The composite is based on the mean of 40-day intervals. The dots

mark the composite anomalies at the 95% confidence level using the *t*-test.

To further analyze the wave dynamics and to better understand the difference between the three types of

DWs and the NDW, the E-P flux anomalies and the E-P flux divergence anomalies are shown in Fig. 8.

The 10-day intervals before the SSW onset and afterward are examined for total planetary waves. For all

types of SSWs, the upward propagation of planetary waves is enhanced before the event onset (Fig. 8a-

d). Comparing the three types of DWs, the E-P flux anomalies are very comparable, and the anomalous

E-P flux convergence center is structured differently (Fig. 8a-c). In both ERA5 and model simulations,



the E-P flux convergence anomalies develop in the entire stratosphere for the type BOTH (Fig. 8a), while

for EA and NA, the convergence anomalies are more concentrated around 50-150 hPa (Fig. 8b, c). The

E-P flux convergence anomalies are weaker for the NDW than all types of DWs, likely due to the weak

E-P flux itself (Fig. 8d).

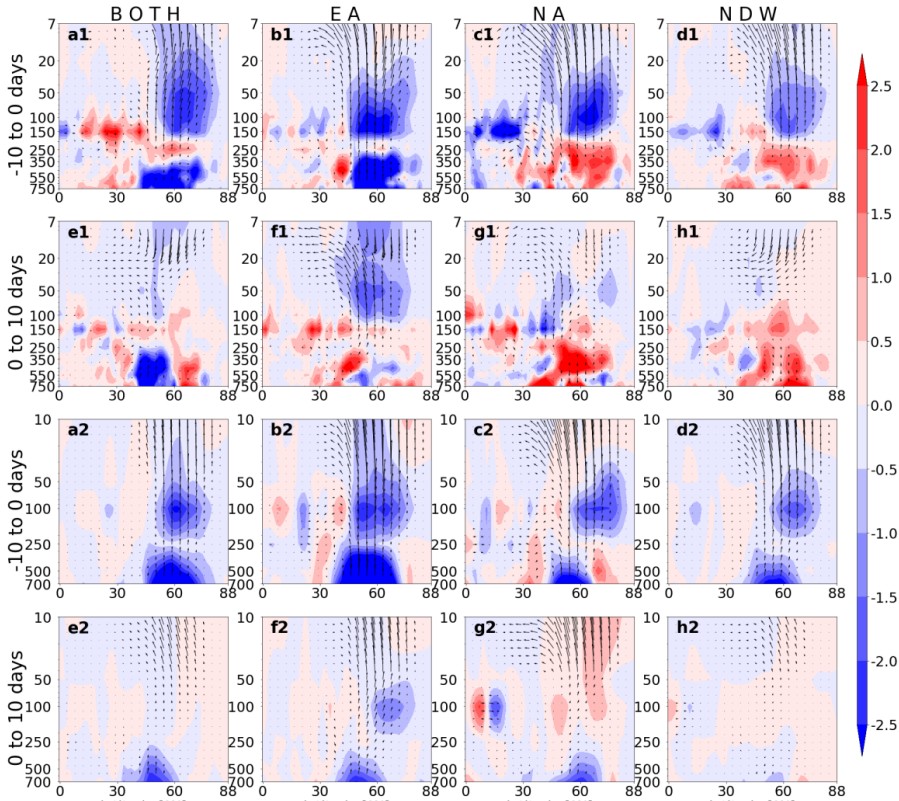

**Figure 8.** Composite E-P flux anomalies (vectors; units: $F_y$ in $10^4$ kg$^3$ s$^{-2}$, $F_z$ in $10^6$ kg$^3$ s$^{-2}$) and E-P flux

divergence anomalies (shadings; units: m s$^{-1}$ d$^{-1}$) by planetary waves (sum of wave 1 and wave 2) for (a,

e) BOTH, (b, f) EA, (c, g) NA and (d, h) NDW events before the SSW onset (top row) and afterward

(bottom row) from two datasets: (a1-h1) ERA5 and (a2-h2) historical simulations by CESM2-WACCM.

The composite is based on the mean of 10-day intervals.

After the SSW onset, the composite 10-day intervals E-P flux anomalies are also not identical for the

three types of SSWs (Fig. 8e-g). The upward propagation in the lower stratosphere and troposphere is

still enhanced for all types of DWs, while it begins to weaken in the extratropical upper stratosphere. As



a consequence, the anomalous E-P flux convergence is still present in the lower stratosphere for all types of DWs. In contrast, weakening of the anomalous E-P flux convergence begins to form for the NDW, and the anomalous downward propagation of waves is also stronger for the NDW than the DW in ERA5,

although model simulations only show E-P flux divergence anomalies in the polar stratosphere (Fig. 8h). The relatively short lifetime of the wave forcing for the NDW likely explains the relatively weak intensity of the stratospheric disturbance and even lacking impact on the near surface in the later period. It is also revealed that the E-P flux divergence (or convergence) anomalies in the upper stratosphere are much weaker than in the lower stratosphere and upper troposphere for all events (Fig. 8e-h).

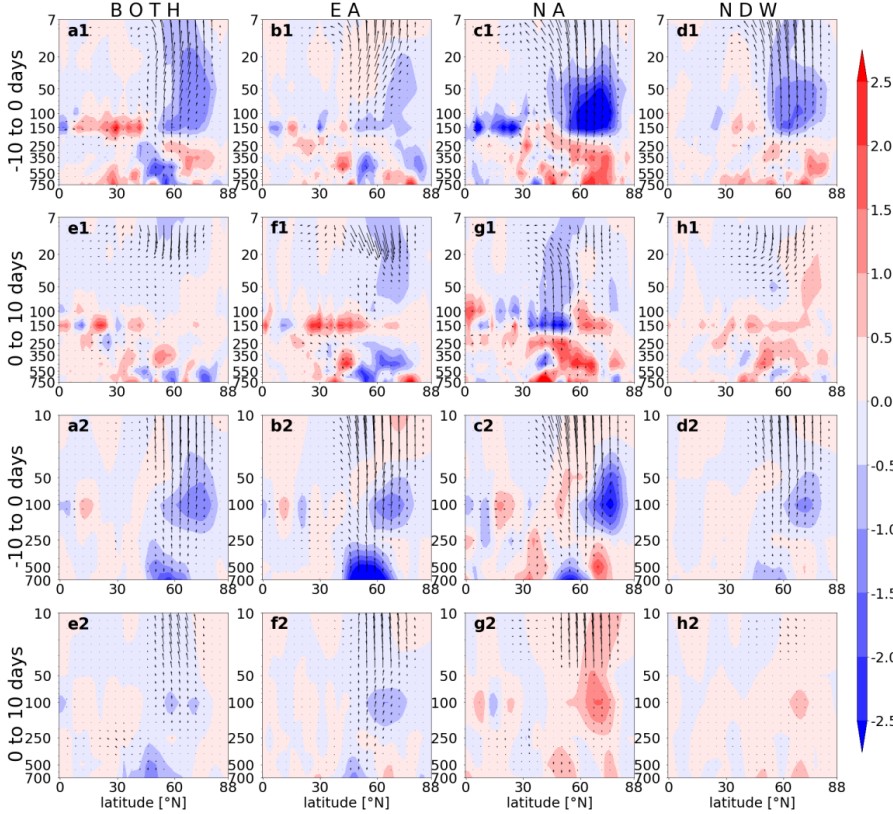


**Figure 9.** Same as in Figure 8, but for planetary wave 1 E-P flux.

The contribution of the wave 1 and wave 2 to the total E-P flux and its divergence is shown in Fig. 9 and Fig. 10, respectively. A decomposition of the waves for E-P flux before the SSW onset can easily reveal

the dynamical difference among the three types of DWs. Specifically, in the pre-SSW onset, the upward



propagation of planetary wave 1 is strengthened for all events in both ERA5 and model simulations, and the E-P flux convergence anomalies by wave 1 also appear for all events (Fig. 9a-d). Moreover, it is obviously observed that the E-P flux convergence anomalies by wave 1 are strongest for NA out of all groups (Fig. 9c).

In the post-SSW 10-day interval period, the upward propagation of wave 1 is suppressed, and the contribution of wave 1 to the deceleration of westerlies (and therefore weakening of the polar vortex) nearly disappears (Fig. 9e-h). Namely, the E-P flux anomalies change the direction from upward to downward, and the E-P flux divergence anomalies are nearly zero. Wave forcing in midlatitudes is still present for DWs in ERA5 or/and model simulations, and the anomalous E-P flux convergence also

persists in the lower stratosphere at midlatitudes (Fig. 9e-g).

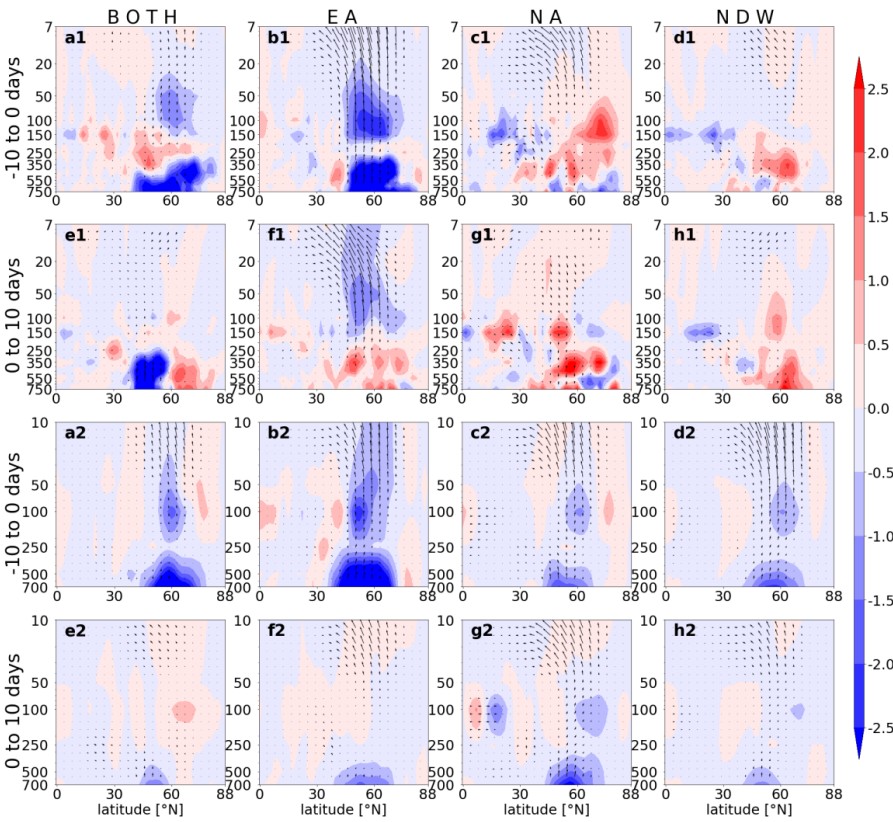

**Figure 10.** Same as in Figure 8, but for planetary wave 2 E-P flux.

Similarly, the dynamics difference for the three types of DWs can be easily seen from the contribution



of wave 2 to the total E-P flux and its convergence. In the pre-SSW periods, wave 2 cooperates with

wave 1 to enhance the upward propagation of planetary waves and therefore the anomalous E-P flux

convergence for BOTH and EA (Fig. 10a, b). Although the wave 2 is enhanced to propagate upward to

the stratosphere for NA and NDW, the E-P flux is little converged, and the wave driven change for the

circulation is not present (Fig. 10c, d). The model simulations generally show a consistent wave 2 E-P

flux anomaly pattern with ERA5.

This difference in the wave 2 forcing also exists in the post-SSW period for the three types of DWs (Fig.

10e-g). Both ERA5 and model simulations show that the vertical component of E-P flux anomalies by

wave 2 reverses the sign for BOTH and NA (Fig. 10e, g), while the upward propagation of wave 2 is still

present for EA (Fig. 10f). As a consequence, the wave 2 forcing for the weakening of the polar vortex

still persists after the SSW onset for EA, while this forcing for BOTH and NA types by wave 2 has

terminated. Changes in the E-P flux anomalies and the divergence anomalies by wave 2 are not noticeable

for NDWs (Fig. 10h).

The wave 1 and wave 2 can exert different effects on the stratospheric polar vortex (Baldwin et al., 2021;

Lindgren and Sheshadri, 2020; Yang et al., 2023). The former propels the polar vortex displaced away

from the Arctic, while the latter elongates and splits the vortex. Previous studies (Anstey et al., 2013;

Kidston et al., 2015; Lehtonen and Karpechko, 2016) found that 2–3 weeks before the displacement SSW,

it is anomalously warmer in the southeastern US and colder in Eurasia. These studies also found 2–3

weeks after the displacement SSW, the southeastern US is anomalously cold while Eurasia is unusually

warm. In the month around the split SSW, the probability of both continents (North America and Eurasia)

being cold at the same time is high. The different percentage of the displacement or split SSWs might

further account for the different circulation and t2m anomalies for different types of DWs.

## 5 Conclusions

SSWs show strong inter-case variability for their impact on the troposphere and near surface. In this

paper, SSWs are classified into different groups based on whether there is downward impact and on

where the downward impact occurs. Using a newly proposed method, this study further classifies 52

SSWs in ERA5 and 273 SSWs in CESM2-WACCM into four groups: DWs with near surface impact

over both Eurasia and North America (BOTH), DWs with impact over Eurasia (EA), DWs with impact

over North America (NA), and NDWs. Finally, 33 DWs (13 + 14 + 6) and 19 NDWs were identified



from ERA5, while 119 DWs (40 + 51 + 28) and 154 NDWs were selected from CESM2-WACCM
simulations. To well distinguish the potential impact diversity of SSWs, the tropospheric circulation
response and the near surface behaviors following these four types of events are revisited in this study.
The main findings in this study are as follows and summarized in Fig. 11.

1) The mean intensity of NDW events in terms of the NAM and the circulation anomalies in the
stratosphere and troposphere are nearly half weaker than DWs events. Comparing the downward impact
of three types of DWs, the persistency of the negative NAM pattern varies with the DW type. On average,
the negative NAM signal in the lower stratosphere can last for >60 days for BOTH and EA, while it only
lasts for ~40 days for NA events. Further, the dipping pattern at the near surface exhibits a continuous
negative NAM signal for BOTH and EA, while it is replaced by positive NAM at both the beginning and
end of the SSW for NA.

2) An isentropic vorticity analysis reveals that anomalously high PV air can move southwards from the
polar stratosphere and enter midlatitudes, thereby affecting the near ground. The cold anomalies over
Eurasia and over North America have precursors in the upstream oceanic regions before the SSW onset.
Anomalously low PV appears over North Atlantic 10-30 days before the EA onset and anomalous high
PV forms over North Atlantic 0-30 days afterward. In contrast, anomalously high PV appears 10-20 days
before the NA onset to 0-10 days afterward over North Pacific – Alaska. The PV anomaly evolutions for
BOTH have a joint characteristic of both EA and NA. Namely, anomalously high PV forms over North
Atlantic and North Pacific – Alaska as the anomalously low PV air enters the Arctic before the SSW
onset for BOTH, while anomalously high PV air sweeps the continents soon afterward. In contrast, the
t2m and 315-K PV anomalies are weaker and more scattered.

3) Adopting relative continental cold anomalies as the criterion for classifying SSWs, changes in
precipitation anomalies are not so sensitive to the DW type as t2m anomalies. Following the onset of all
types of SSWs, the rainfall band shows a southward shift especially over the Atlantic–Europe sector,
exhibiting a rainfall anomaly dipole in the Western Hemisphere. Namely, the precipitation in subtropical
Atlantic – Mediterranean Sea increases, while precipitation in high-latitude regions decreases. Previous
studies have found that the southward shift of the precipitation band is associated with an equatorward
shift in the storm track, and the intensification of moving cyclones at low latitudes (e.g., Thompson and
Wallace, 2001; Huang and Xie, 2015).

4) The dynamical processes during DWs and the NDW are various before and after the SSW onset. In



the pre-onset period, the negative NAM has been shaped at surface for BOTH and EA, while weak NAM

still dominates at surface for NA, although anomalous high begins to form over the Arctic. In the post-

onset period, the anomalously low band in midlatitudes shows different structures for DWs. Three low

centers are clearly present over Western Europe, East Asia, and northern North America especially in the

lower stratosphere for BOTH, while only the low center over East Asia still exists for EA and NA. The

near surface NAM pattern is well organized with the midlatitude negative pressure anomalies over sea

larger than over lands for BOTH and EA, while the pressure anomalies are larger over lands than over

oceans for NA. For the NDW, the circulation anomaly amplitude is only half or even one third of that for

DWs.

5) The dynamic differences for the three types of DWs are also featured by the wave activities. Firstly,

the wave 1 forcing for them (BOTH, EA, and NA) shows a similar structure in the pre-onset period: the

upward propagation of wave 1 increases, and the anomalous E-P flux convergence denotes a dissipation

of wave 1 in the lower stratosphere. In the post-onset period, the enhancement of upward-travelling wave

1 terminates for BOTH and EA, although a small enhancement is still present for NA. Secondly, the wave

2 forcing behaves in very different spatiotemporal structures: the wave 2 is enhanced to propagate upward

in the pre-onset period for all types of DWs, but the wave dissipation in the stratosphere is only detected

for BOTH and EA. In the post-onset period, the upward propagation of wave 2 is instantly suppressed

for BOTH and NA, while the wave 2 forcing is still strong for EA.

6) Considering that displacement and split events have different impacts on the two continents (Anstey

et al., 2013; Kidston et al., 2015; Lehtonen and Karpechko, 2016), a larger proportion of displacement

SSW for NDWs and NA might explain their weak downward impact on near surface especially over

Eurasia.



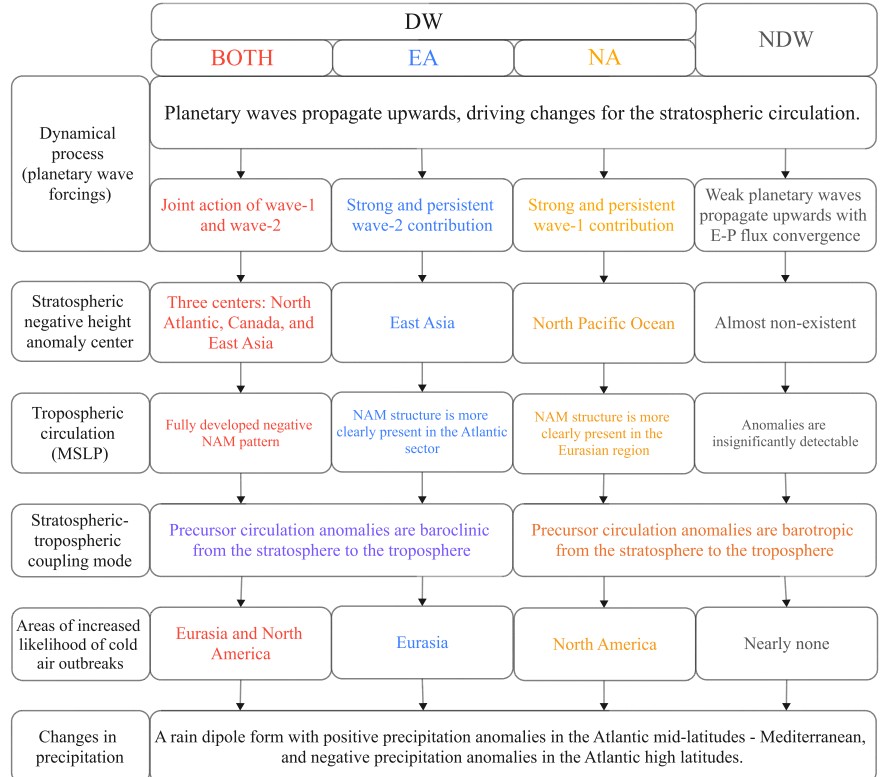

**Figure 11.** Schematic charts comparing the three types of DWs and the NDW. Aspects of comparison include preceding wave forcings by wave 1 and 2, the circulation pattern in the stratosphere, the stratosphere-troposphere coupling, and the downward impact on near surface.


Compared with previous studies (Anstey et al., 2013; Domeisen et al., 2020), our study reveals the diversity of the DW events and distinguishes the potential impact on both continents in the Northern Hemisphere. The findings underscore the significant regional diversity in the tropospheric response to different SSW types, with distinct durations, intensities, and spatial patterns. These new insights enhance

our understanding of the mechanisms driving surface impacts and could improve our understanding of weather and climate variability associated with SSWs. This consideration is reasonable, because cold air outbreaks on both continents are not in pace most of the time (Butler et al., 2017; Yu et al., 2024). White et al. (2019) found that at least 35 events should be present to get robust differences between DW and NDW events. The sample sizes from CESM2-WACCM generally meet the case requirement in this study.

Using more samples from CMIP6 multimodel outputs, a deeper understanding of different types of DW



events is possible, left for future investigation. Further, the distribution of various types of events (see Table 1) shows strong interdecadal variability in past decades. Whether this change is an internal climate variability or forced by global warming due to anthropogenic emission is still unknown, worth exploring in the future.

**Data availability**

The ERA5 reanalysis is available from the ECMWF (https://cds.climate.copernicus.eu/cdsapp#!/dataset/reanalysis-era5-pressure-levels-monthly-means?tab=form). The CESM2-WACCM historical simulations participating CMIP6 are available from the ESGF (https://esgf-node.llnl.gov/projects/cmip6/).

**Author contributions**

RL and JR designed this research. RL and JR analyzed the data. RL provided the data analysis methods. RL wrote the manuscript draft. JR reviewed and edited the manuscript.

**Competing interests**

The contact author has declared that none of the authors has any competing interests.

**Acknowledgments**

The authors express their gratitude to the National Natural Science Foundation of China for the funding support. The authors thank the High Performance Computing Center of Nanjing University of Information Science & Technology for their support of this work. The ECMWF is acknowledged by the authors for the accessible modern reanalysis data.

**Funding**

This research was supported by the National Natural Science Foundation of China (Grant nos. 42361144843, 42322503, 42175069, and 42288101) and the Qinglan project of Jiangsu.

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
