# Peer review of "Sorting sudden stratospheric warmings with the downward tropospheric influence using ERA5 and CESM2-WACCM"

_EGUsphere, 2025_

## Referee Comment (RC2)

Review of "*Sorting sudden stratospheric warmings with the downward tropospheric influence using ERA5 and CESM2-WACCM*" by Lu and Rao for Atmospheric Chemistry and Physics

**Summary:**

This study analyzed different types of sudden stratospheric warming (SSW) events based on their downward impacts. Besides only classifying SSW events into those with (DW) and without (NDW) downward impacts, the authors further classified the downward events into three different subgroups based on the region where the cold air outbreaks occurred and looked into the differences in various dynamical quantities for different types of DW events. The study provides an interesting aspect to further study the downward impacts of SSWs and documents different behaviors for four types of SSWs. The paper is well-written in terms of language and structure. However, there are two main concerns listed in the major comments below regarding the analysis and description of the results.

Overall, this study is interesting and meaningful for its detailed examination of DW events and their impacts on various regions. However, the analysis is somewhat shallow, and some of the descriptions and interpretations are not accurate or clear. I suggest a major revision before publication.

**Major comments:**

1. The authors want to use the CESM-WACCM data to support the findings obtained from the ERA5 data, given the small sample size of SSWs in ERA5. However, the authors described it too generally for the analysis using the CESM-WACCM model data in many places in the paper, by only saying that the model results are consistent with or resemble the ERA5, which, in many places, is not true or accurate (see specific comments below). There are quite some differences in the analyses between ERA5 and model data, which could raise questions and concerns about the robustness of the results found in ERA5 (although the model is not perfect and has its own biases). I don't think the authors discuss the differences and their implications carefully enough.

2. There is no in-depth analysis beyond purely describing the figures, which can, in the authors' words, "enhance our understanding of the mechanisms driving surface impacts". For the discussion in almost all figures, the authors described the differences in the dynamical variables between the three types of DWs in detail, but without explaining how these patterns or unique processes play a role in the development of the three types of DWs. In fact, it is not even clear whether these patterns are critical for the cold air outbreaks in different regions. Besides, some descriptions of the figures are not accurate or consistent with what the figure shows (see specific comments below).

**Specific comments:**

L110-111 and L112-113: The introduction of the CESM2-WACCM model simulation for the SSWs and downward impacts is too general. It would help to add a bit more detail. For example,

does it simulate SSWs (and the downward impacts) well in terms of frequency or magnitude or duration/persistence? Is CESM2-WACCM a high-top model? What is its vertical resolution?

L153: What is the definition of the onset of DWs?

L155-160: The regions for NA and EA are too broad. Within Eurasia and North America, it is not necessary that the entire region experiences the cold air outbreak after SSWs. For example, very often it could be some specific part of the US that experiences extreme cold weather. Taking the average for such a large region could weaken the cold responses. It is also not clear about the classification of the event types. When an SSW event is classified into the EA type, does it mean that both Asia and Europe meet the criteria? Or is only one region (either Asia or Europe) meeting the criteria good enough and counted?

L161: How did the authors decide the -0.3℃ here? Does it correspond to a certain standard deviation or percentile threshold? Besides, -0.3℃ seems relatively small, could averaging over such a large region as mentioned above make the overall anomalies small?

L211 and the entire discussion related to Figure 1: Are these differences statistically significant across these three types of DWs? The same questions apply to all other analyses shown below.

L212-L217: How do these different behaviors before SSWs determine, influence, or play a role in different types of downward impacts after SSWs?

L225-230: Are the NAM anomaly contrasts before and after events for each type of SSW event robust? For example, the contrast in the NA events is not shown in the model. I understand that the model is not perfect and has its own biases. However, since the sample size in ERA5 is also small while the model has more samples, it could also be that the features in ERA5 are not robust and due to a specific individual strong event.

Besides, are the differences in these contrasts for different types of SSWs statistically significant? If the contrast is important, how does it play a role in influencing the development of different types of SSW events?

L264-265: I agree that the authors can interpret that the main features in the PDF of the model are similar to those of ERA5. However, there are also some differences that may require a second thought. For example, we can see the shift of the peak in EA and a higher probability with smaller negative NAM values of EA in the model compared to ERA5, and a clearer peak and the overall shape of the PDF of NA in the model differ from ERA5. This would make me wonder about the robustness of the features found in the ERA5. For instance, is the difference in the mean or median between BOTH and EA actually significant or distinguishable, because it seems not to be the case in the CESM2-WACCM model (the PDFs of BOTH and EA have larger overlap in the model than in ERA5)?

L300-301: It seems like a seesaw pattern in t2m between Eurasia and North America, which is interesting. Is this a firm and robust relationship in which, when one region is anomalously cold, the other region is anomalously warm? If so, why and what causes this seesaw pattern?

L314-317: First of all, the differences between ERA5 and the model are not minor, especially from day -40 to day -10. The t2m anomalies are almost zero in this time period in the model. This again goes back to my previous concern about the robustness of the results in ERA5 and brings to the next question: do the t2m anomalies before SSWs precondition the cold anomalies after SSWs? This analysis cannot really show or support the role or importance of the t2m anomalies before SSWs in developing BOTH DWs. The same questions for other types of DWs.

L322-323: Does the evolution of t2m here really relate to or result from SSWs for regions of Europe and Asia, given that the anomalies are not statistically significant and the t2m anomalies are almost unchanged during the period in Europe?

L355-356: It seems like the high IPV center is consistently over Europe throughout the whole period. There is no clear movement of high IPV from the figure, and there is a sudden strengthening of the anomalies, which indicates the IPV does not seem to be conserved but is under external forcing. Thus, it cannot tell the source of the IPV based on the movement of high IPV.

Figure 7 and the relevant description: After reading the whole description of Figure 7, it is good to see different patterns in different types of DWs and NDW, but it is still not clear to me what dynamical processes or what and how these features and patterns before the SSWs influence the development of different types of DWs. The analysis is just descriptive, not really in-depth enough to reveal the differences in the underlying dynamics.

L480-481: From the vectors in figure 8e1 and g1, they seem to be pointing downward over the high latitudes, which indicates the opposite of what the authors describe here. Besides, the convergence is very weak in the stratosphere for BOTH and NA. Are the anomalies here statistically significant?

L486-487: How does the short lifetime of the wave forcing explain the lack of impacts on the surface?

L503-505: The description is not accurate. Only Figure 9g1 shows this. The differences between the model and ERA5 are quite dominant.

L512-514: There is a divergence in the lower stratosphere for the NA DW. Could the author provide any explanation for this? How does this feature lead to or play a role in the NA surface impact? Please explain this.

L514-515: I disagree. The differences between the model and ERA5 are quite dominant to me.

L517: "Both ERA5 and model simulations" Again, I think the differences between the model and ERA5 are large. For example, the model does not show the EP flux convergence anomalies for EA.

L517-518: The arrows of the vectors are hard to see. This is also the question for Figures 8 and 9. In fact, the authors did not mention whether there is any scaling when plotting the EP flux, which is usually the convention. The small magnitude of the vectors indicates the anomalies are small. Are the EP flux anomalies statistically significant?

Discussion for Figures 8-10: I can see the differences in the EP flux and EP flux divergence across the three types of DWs. However, I think there is a lack of in-depth analysis of how these differences influence/determine the types of DWs. For example, after reading the whole section, I still do not know how the anomalous upward propagation of wave-2 wave activity after SSWs leads to the coldness over EA. Are the displacement or split of the vortex more important, or the wave activity itself more important?

Figure 11 and its summary: It is clear to summarize the main differences using this schematic. However, there is a lack of connection between all these dynamical processes and wave activities flux, and coupling, which lead to or contribute to the final development of different types of DWs. There is no clear interpretation to put all these components into a thread, which makes the study purely descriptive and lacks in-depth insights.

L599-600: I don't see any improved understanding of the mechanisms. Please see my comments above.

**Minor comments:**

L155 and Table 2: It will help to summarize the percentage of each type of event for both ERA5 and the model.

L164: It would help to briefly describe the method by Maycock and Hitchcock (2015) so that it would be clearer when you mention the parameters you modified below.

L238: What is Fig. 1e1 and f1?

L240: I agree there is nice consistency between the model and ERA5 in most cases here. However, the magnitude of the negative NAM is apparently weaker from the lower stratosphere to the mid-troposphere in the model than in ERA5. Something like this can be mentioned in the section where you introduce the overall model performance in downward impacts.

L249: I don't think the authors introduced how to compute or obtain the NAO index before.

L254-255: How can DWs modify the PDF of the NAO?

L291-292: Keep in mind that the t2m anomaly pattern contrastingly different for different types of events is not something new because this is how you define the three types of DWs and NDW.

L329-330: Figure 4b2 does not fully support this statement for Europe.

L331-332: The Europe region in Figure 4c2 does not support the description.

L333-334: This sentence is confusing. I don't understand which features or lines in the figure the authors talked about.

L442: Isn't the 100 hPa height anomaly over the North Atlantic also negative?

L525-529: Isn't the finding opposite to the previous studies?

---

## Author Comment (AC1)

**Response to Reviewer # 1**

Review for "Sorting sudden stratospheric warmings with the downward tropospheric influence using ERA5 and CESM2-WACCM" by Lu and Rao.

**Summary**
This study investigates how different types of downward-propagating sudden stratospheric warming (SSW) events are associated with distinct regional surface cold extremes and classifies them based on their surface impacts, using reanalysis data and CESM-WACCM simulations. According to the abstract, introduction and conclusion, the paper aims not only to classify cases of surface cold extremes associated with SSWs, but also to explain the mechanisms behind how different types of surface cold extremes are caused by difference in SSWs. To this end, the study presents a comprehensive analysis using both observations and CESM-WACCM. One of the main contributions of the paper is the classification of downward-propagating SSW events based on the regional characteristics of the associated surface cold extremes, followed by a global analysis of the related dynamical fields.

Response: Thank you for your positive comments. This study is motivated by the variety of the surface anomalies following the SSWs. As we introduced in the first section of the paper, not all downward propagating SSWs have a cold impact on NA or EA. This classification is different from those in literature, but it really improves our understanding of the SSW.

Firstly, The wave forcings for the SSWs are rich and various: BOTH type is forced by comparable strong wave 1 and wave 2 pulse, EA type is forced by strong wave 2 pulse (EP flux convergence is much larger for wave 2 than wave 1), NA type is forced by strong wave 1 pulse (EP flux convergence is much larger for wave 1 than wave 2), and NDW is associated with the strong total waves (while wave 1 and wave 2 are much weaker for NDW than for DW). This dynamic difference is very clearly shown in Figure 8-10.

Second, the evolution of the circulation during SSWs are diverse: Following the SSW onset, BOTH type is charactered by a complete NAM pattern with the negative height anomaly lobe in midlatitudes extending in a zonal band encircling the earth. The EA type is feature with a NAM pattern but the negative lobe in midlatitude split into two lobes. The NA type is accompanied with the polar anomalous high with the midlatitude negative height anomaly lobe missing. The NDW type has a much weaker NAM like circulation.

Third, as a consequence, the near surface temperature response is diverse. This diversity is not just due to the classification definition, which is consistent with the atmospheric circulation anomalies.

However, the main analyses are mostly descriptive, such as "this type tends to have these features," without sufficient efforts to clarify the causal connections between processes. This creates a mismatch between the intended positioning of the paper and the actual description of the results, which may confuse readers about the central message of the paper. Moreover, even if the main purpose was the classification of SSW types, there are places where the descriptions overinterpret differences between the types that may naturally result from the way they are classified.

Response: Thank you for your criticism comments. During the revision, we considered all of your comments and made corresponding revision.

First, we provide as accurate description as possible and made sufficient effort as possible to clarify the causal connection between the preceding wave forcing, the SSW, the circulation changes, and the near surface response. This chain is finally provided in Figure 11.

Second, all of the mismatch between the results and the explanation has been revised and clarified. Places that might mislead our reviewer and readers have been removed or change with suitable words.

Third, the descriptions overinterpret differences between the types is revised. We also added the difference between CESM2-WACCM and ERA5 in the revision this time.

The reviewer has a viewpoint that the difference is sourced from the way SSWs are classified. We do not deny this. The core of this study is why we classify the SSW according to the surface response, which is what we emphasizes. Thank you!

Nevertheless, I believe the paper has potential, since the analyses are extensive and well-organized. It would be publishable as a descriptive classification study if the authors revise the manuscript to focus on the meaningful dynamical differences revealed by the classification, while excluding the differences that naturally arise from the way the types are defined.

Response: Thank you for your comments. The dynamics differences are consistent with our classification. The near surface difference is the motivation of this study, and we analyzed the difference from several aspects: the wave forcings preceding the SSW, the SSW associated circulation (the NAM pattern structure), the evolution of the downward propagation of the NAM, and the near surface.

All the chains are analyzed and summarized in Figure 11, and all of those results are consistent. We describe our results and give explanation as accurate as possible this time.

**Major Comments**
**Unclear objective: classification or mechanism?**

If the goal is to understand the cause of different SSW surface impacts, the paper needs to move beyond descriptive comparisons and provide a clearer explanation of how variables interact across steps. If the goal is to classify the types, the paper should clarify up front that it is intended as a classification study based on observed differences.

Response: The primary objective of this paper is to classify SSWs based on different tropospheric responses and to conduct an investigation into the underlying mechanisms from the dynamic difference.

To well address your concern, we have incorporated this additional text into the manuscript.
· "The primary objective of this paper is to classify SSWs based on different tropospheric responses and to conduct an investigation into the underlying mechanisms from the dynamic differences." (L92-93).

Furthermore, we have expanded upon the discussion of physical mechanisms within the paper.

**Insufficient interpretation and unclear physical connections**
The authors should avoid wording that may be misinterpreted as implying a causal relationship between features that simply co-occur. It would be helpful to clarify which parts of the discussion are supported by physical reasoning and which are more descriptive.
Response: We have avoided wording that might mislead our reviewer and readers as accurate as possible. Features that cooccur are consistent. The physical reasoning is mainly shown in the dynamics analysis section (Section 4). The description of the difference is mainly shown in the tropospheric response comparison section (Section 3).

1.  Some differences reported in the results seem to directly stem from how the cases were defined, but this point is not acknowledged. For example, the EA-type cold extremes are defined to cover a broader longitudinal range. Since the NAM index more reflects zonal-mean changes, it is not surprising that EA cases show a stronger and persistent NAM signal than NA cases. This might be a natural result of the classification.
Response: The EA cold appears in EA and BOTH types, and the NA cold appears in NA and BOTH types. Both continents are compared. Most of the composite is significant. The composite t2m anomalies in Figure 3 are significant, which might imply that our classification is meaningful rather than noisy only.

If the classification lacks this "natural" result, the classification fails. In climate study, those natural results can be broadly seen. We defined the ENSO index, so the composite is a ENSO pattern. We define the QBO index, so the composite is a QBO pattern. We

classify the ENSO types, so we get the CP and EP patterns … … All those are predefined, and the dynamics are consistent with this classification. The classification is a useful tool to understand the diversity of our real world.

After careful consideration, we insist on that our classification is meaningful, physically distinguishable, and helpful to understand the SSW diversity.

2. The paper tends to overinterpret co-occurring patterns as physical links. The paper lists many local differences across types but does not sufficiently explain whether these patterns are meaningfully connected.

Response: In our analysis of the geopotential height field, we linked various anomalies through circulation changes. The circulation field provides a more intuitive view of the characteristics of the NAM pattern. The near surface change follows the circulation change on the subseasonal timescale. Furthermore, anomalies in the circulation field, particularly the movement of the polar vortex, can be correlated with changes in surface temperatures before and after various events.

3. the paper lists a number of detailed local differences between SSW types and presents them as if they are physically meaningful, and this seems to depend on the statistical significance of the composite. However, it is not always evident whether these differences are statistically significant. There are figures which show that composite for NDW show widespread dotted areas of statistically significant values (e.g. Fig. 6d), and the statistical testing method is not clearly explained. This raises the concern that some of these apparent differences may not be meaningful physical distinctions, especially in relation to the difference in SSW downward impact.

Response: The significance test used in this paper is the t-test, which test if the resample mean is different from the total mean or if the means of two resamples are different. In some figures, specific regions within the NDW plots do indeed pass the significance test, but these regions are markedly smaller and much scattered than the significant areas observed in the DW types.

To well address your concern, we replotted Figure 6 and give a stricter significant test. The significant area for the NDWs is much limited in ERA5 and CESM2-WACCM after revision. The test method is mentioned in the caption of Figure 6.

4. The anomalies suggested as precursors are not clearly distinguished as being part of the SSW-related signal or independent tropospheric variability.

Response: Prior to the occurrence of the SSW, significant changes have already taken place in the atmospheric circulations, which drive the onset of SSWs. It is too controversial to say this change is part of the SSW or the tropospheric variability. Since they are mixed together and indistinguishable, to the best of our knowledge. Despite this, we still believe that these precursor anomalies are intrinsically linked to the SSW.

**Methodology**

1. Even if the main objective of the study was the type classification, more careful methodological design would be necessary. For example, EA DW cases already show strong cold anomalies before the SSW occurred, which raises questions about whether the cold extremes are truly caused by SSW. If the anomalies are computed based on a fixed climatology, long-term signals may be included. However, there is also no clear method for determining whether post-SSW cold events are actually caused by the SSW. This makes it difficult to rule out the possibility that some tropospheric anomalies (or anomalies of longer timescale) developed independently of the SSW.

Response: Thank you for your suggestion. We agree with your viewpoint. Regarding the significant cold anomaly response observed at the surface following SSW events, many studies have been conducted within the academic community, although the controversy still exists whether the stratospheric and tropospheric variability can be split. Since the climate is a nonlinear system, we do not think it is easy to distinguish the stratospheric and tropospheric variability, which is far beyond the scope of our study.

In our study, through analysis of the circulation (Figure 7), we show that the polar vortex has already begun to weaken and deviated from the pole before the SSW occurrence. The region towards which the polar vortex migrated coincides with the area where surface cold anomalies emerged before the event. Therefore, we conclude that although the timing of the surface cold anomaly was earlier, the stratospheric influence can not be ignored.

2. How CESM-WACCM contributes to the overall interpretation is not clear. Is it simply to provide more cases, or to test mechanisms? For example, when CESM-WACCM and observational results differ, the physical meaning of the differences is not discussed. Although the study analyzes both ERA5 and CESM-WACCM, the interpretation is mostly based on ERA5.

Response: At the very beginning of our paper, we only show the ERA5 reanalysis, which has really limited sample sizes. As our reviewers suggest in the last round review, we provide the CESM2-WACCM results. We hope to find a balance between the reviewers. We wish our reviewer can understand us.

The primary reason for using the CESM-WACCM model was to validate the characteristics identified in ERA5. The two datasets exhibited consistent behaviours in key findings and conclusions, although minor differences exist. In the revised version, we have expanded our discussion of the differences between the two datasets. After all, no models are perfect.

**Lack of structural and editorial refinement**

1. Although the analysis is limited to the Northern Hemisphere, this is not explicitly stated in the methodology.

Response: Thank you for your suggestion. We have provided additional explanations in the methodology section. (L122)

2. The structure of the methodology section also lacks consistency. For example, the CESM-WACCM model description is under the subsection "2.1 Reanalysis Data."

Response: Thank you for your suggestion. Adjustments have been made. (L109)

3. While the study defines four geographical domains (Europe, Asia, the United States, and Canada) the classification later becomes just two types (NA and EA), without explanation of the criteria used to group these regions. Furthermore, there is no explanation for why those particular domains were chosen for each region.

Response: One of the main purposes of the article is to distinguish SSW events with different impacts on two continents (North America and Eurasia). However, a single SSW will only have an impact on a portion of the mainland, making it difficult to cover the entire continent most of the circumstances.

For this reason and as the other reviewer suggests, we further divide the mainland. We choose these specific areas as representatives because cold tends to appear in those subregions following the SSWs. We made more explanation this time:

· "Considering that cold surges are more active over the Eurasia (Europe + Asia) and North America (US + Canada) following the DWs, a comparison between the cold anomalies over the two regions can further classify the DWs into three types…" (L159-161)

· "For the mainland, as long as one district meets the criteria, it is considered to be associated with the downward impact of the preexisting DW." (L168-169)

4. A few grammatical errors are present in the manuscript.

Response: Inspections and modifications have been made.

---

## Author Comment (AC2)

**Response to Reviewer # 2**

Review for "Sorting sudden stratospheric warmings with the downward tropospheric influence using ERA5 and CESM2-WACCM" by Lu and Rao.

**Summary**

This study analyzed different types of sudden stratospheric warming (SSW) events based on their downward impacts. Besides only classifying SSW events into those with (DW) and without (NDW) downward impacts, the authors further classified the downward events into three different subgroups based on the region where the cold air outbreaks occurred and looked into the differences in various dynamical quantities for different types of DW events. The study provides an interesting aspect to further study the downward impacts of SSWs and documents different behaviors for four types of SSWs. The paper is well-written in terms of language and structure. However, there are two main concerns listed in the major comments below regarding the analysis and description of the results.

Overall, this study is interesting and meaningful for its detailed examination of DW events and their impacts on various regions. However, the analysis is somewhat shallow, and some of the descriptions and interpretations are not accurate or clear. I suggest a major revision before publication.

Response: Thank you for your criticism comments. We have made corresponding revision at this time.

**Major comments**:

1. The authors want to use the CESM-WACCM data to support the findings obtained from the ERA5 data, given the small sample size of SSWs in ERA5. However, the authors described it too generally for the analysis using the CESM-WACCM model data in many places in the paper, by only saying that the model results are consistent with or resemble the ERA5, which, in many places, is not true or accurate (see specific comments below). There are quite some differences in the analyses between ERA5 and model data, which could raise questions and concerns about the robustness of the results found in ERA5 (although the model is not perfect and has its own biases). I don't think the authors discuss the differences and their implications carefully enough.

Response: We agree that we did not mention the results based on the CESM2-WACCM directly. On one hand, we indeed used CESM2-WACCM historical outputs to increase the sample size. On the other hand, the observed featured from ERA5 can be confirmed. The reviewer thought that the discussion on the difference between CESM2-WACCM and ERA5 is too weak, because we focus on the robust and similar features of CESM2-WACCM and ERA5.

In order to well address your concern, we add discussions in the paper. In the revision, we conducted a more detailed examination of the discrepancies between the two datasets (L282-286, 513-515, 562-563).

2. There is no in-depth analysis beyond purely describing the figures, which can, in the authors' words, "enhance our understanding of the mechanisms driving surface impacts". For the discussion in almost all figures, the authors described the differences in the dynamical variables between the three types of DWs in detail, but without explaining how these patterns or unique processes play a role in the development of the three types of DWs. In fact, it is not even clear whether these patterns are critical for the cold air outbreaks in different regions. Besides, some descriptions of the figures are not accurate or consistent with what the figure shows (see specific comments below).

Response: We have incorporated a more detailed analysis of dynamics in the result and the conclusions section. Since most of specific comments are raised around the two major comments, we provided a more detailed reply below.

The difference between the DWs and among DWs versus NDWs is robust in the two datasets. The dynamic difference has appeared before the SSW onset, which enrich the diversity of the SSW, and lead to different consequences in the troposphere and near surface. This is the original motivation of this study. The wave forcing difference before the SSW results in different behavior of the stratospheric polar vortex, which is discussed in Figures 7-10. We also summarize our main finding in Figure 11.

**Specific comments**:

L110-111 and L112-113: The introduction of the CESM2-WACCM model simulation for the SSWs and downward impacts is too general. It would help to add a bit more detail. For example, does it simulate SSWs (and the downward impacts) well in terms of frequency or magnitude or duration/persistence? Is CESM2-WACCM a high-top model? What is its vertical resolution?

Response: This model has been assessed in existing literature (Liang et al., 2022), so we did not mention too much of those details. We are sorry if we cause trouble for our reviewer.

To well address your concern, we provide more detail this time.

· "The CESM2-WACCM model has been shown to well simulate the SSW frequency, displacement versus split types, and the downward impact (e.g., Liang et al., 2022). The historical simulation starts from 1850 and ends in 2014, which is one of the common experiments from CMIP6. Daily outputs from CESM2-WACCM have a horizontal resolution of 0.9° latitude by 1.25° longitude at 8 standard pressure levels from 1000 to 10 hPa." (L112-116)

Actually, the downward propagation of the SSWs has also been shown in Figure 1.

L153: What is the definition of the onset of DWs?

Response: DWs are SSWs. Their onset is defined as the first day when the u-wind is reversed (L124-125)

L155-160: The regions for NA and EA are too broad. Within Eurasia and North America, it is not necessary that the entire region experiences the cold air outbreak after SSWs. For example, very often it could be some specific part of the US that experiences extreme cold weather. Taking the average for such a large region could weaken the cold responses. It is also not clear about the classification of the event types. When an SSW event is classified into the EA type, does it mean that both Asia and Europe meet the criteria? Or is only one region (either Asia or Europe) meeting the criteria good enough and counted?

Response: We agree with your viewpoint. Actually, a broad average will lead to a bias for some local regions with cold extreme if the entire area does not experience cold.

In order to well address this concern from our reviewer, we have divided the continent into smaller regions. We have also noticed the issue that taking the average value within the region weakens the intensity of the cold response. Therefore, we have fully considered this point (including the proportion of days and temperature threshold) in the criteria for determining whether a cold event occurs.

· "Within one region, the area-averaged temperature anomalies over a 40-day period are considered to be associated with the downward impact of the DW if the anomalies meet the following two criteria: 1) the percentage of days with cold anomalies is greater than 50%; 2) the mean temperature anomaly over a 40-day period is less than -0.3°C. For the mainland, as long as one district meets the criteria, it is considered to be associated with the downward impact of the preexisting DW." (L168-169)

L161: How did the authors decide the -0.3℃ here? Does it correspond to a certain standard deviation or percentile threshold? Besides, -0.3℃ seems relatively small, could averaging over such a large region as mentioned above make the overall anomalies small?

Response: When determining the threshold, we took into account the possible weakening effect of continent scale average. At the very beginning we tried different criteria, and the composite pattern is similar but the anomaly amplitude is different. In general, as the threshold absolute value is larger, the composite anomalies are stronger. The value of -0.3℃ was selected as the most appropriate one through analyzing the cold anomalies in the two continents after each SSW event. This threshold can classify all DWs into the three types. The threshold any larger will miss some DW cases in the three types: BOTH, NA, and EA.

Revision was made in line 164.

L211 and the entire discussion related to Figure 1: Are these differences statistically significant across these three types of DWs? The same questions apply to all other analyses shown below.

Response: We consider your suggestion. The cross difference will lead to too many plots if we produce those differences. We provide the difference plots in Fig. R1 exclusively for your reference.

You might find from Figure R1 that the difference between BOTH and NDW, and the difference between EA and NA are significant in wide areas.

To well address your concern, we also added some revision.
· "…while it is very short in the persistent time and show a low significance level for NA." (L249)
· "The difference between BOTH and NDW and between NA and EA is most significant in the troposphere and near surface, implying a diversity in the persistency and NAM intensity among SSW types." (L253)

[Figure]

**Figure R1.** Same as in Figure 1, but for difference between (a) BOTH and NDW, (b) NA and EA. The top is shown for ERA5 and bottom is shown for CESM2-WACCM.

L212-L217: How do these different behaviors before SSWs determine, influence, or play a role in different types of downward impacts after SSWs?
Response: The different characteristics of NAM before SSWs are consistent with different changes in the overall circulation. For example, the low-level circulation for BOTH and EA is continuous, while the low-level circulation in NA has not changed much. (L248-259)

L225-230: Are the NAM anomaly contrasts before and after events for each type of SSW event robust? For example, the contrast in the NA events is not shown in the model. I understand that the model is not perfect and has its own biases. However, since the sample size in ERA5 is also small while the model has more samples, it could also be that the features in ERA5 are not robust and due to a specific individual strong event.

Response: We agree with you that the NAM anomaly contrast before and after events might be due to a small sample size. We added some discussion this time.

· "It evolves from weakly positive to weakly but persistently negative for type EA in ERA5, and this signal transition is not significant in the model. The strong phase transition for EA in ERA5 may be due to the small sample size and the influence of individual cases." (L242-245)

Besides, are the differences in these contrasts for different types of SSWs statistically significant? If the contrast is important, how does it play a role in influencing the development of different types of SSW events?

Response: For differences in comparisons between various SSW events, please refer to Fig. R1. SSW diversity is characterized by the preceding wave forcing and the consistent NAM evolution as we state in the paper. The corresponding conclusion is shown in Figure 11.

L264-265: I agree that the authors can interpret that the main features in the PDF of the model are similar to those of ERA5. However, there are also some differences that may require a second thought. For example, we can see the shift of the peak in EA and a higher probability with smaller negative NAM values of EA in the model compared to ERA5, and a clearer peak and the overall shape of the PDF of NA in the model differ from ERA5. This would make me wonder about the robustness of the features found in the ERA5. For instance, is the difference in the mean or median between BOTH and EA actually significant or distinguishable, because it seems not to be the case in the CESM2-WACCM model (the PDFs of BOTH and EA have larger overlap in the model than in ERA5)?

Response: Thank you for your suggestion. Difference indeed exists between the ERA5 reanalysis and CESM2-WACCM. However, those differences are very minor when they are compared with the consistent part of the results from CESM2-WACCM and ERA5. We also agree with the reviewer that no model is perfect, but the model we used provide a more robust support for the diagnosis for the ERA5 reanalysis. The inclusion of a model in our paper is suggested by the reviewers in the last round review.

Our reviewer emphasizes the difference between CESM2-WACCM and ERA5 in many places of the review report, but this difference is not the core of the paper. This paper is not aimed to assess the skill of the CESM2-WACCM, since this model has a relatively good skill in the stratospheric circulation. The difference between the CESM2-WACCM and ERA5 is probably the bias, and can also be due to the limited samples in ERA5. However, the consistent part of CESM2-WACCM and ERA5 is still

seen. BOTH type shows a PDF distribution to the left, followed by EA and NA. This is the key point and what we emphasize.

To well address your concern, we have also revised the corresponding sentences:
·  "Namely, the mean probability density function for BOTH and EA is much more left-skewed than that for NA. Consistent with ERA5, the probability density functions of BOTH and EA exhibit a higher degree of overlap. The probability density function for NA is distributed more flattened than that for BOTH and EA. Moreover, due to the larger sample size, the peak values can be seen more clearly for NA." (L282-286)

L300-301: It seems like a seesaw pattern in t2m between Eurasia and North America, which is interesting. Is this a firm and robust relationship in which, when one region is anomalously cold, the other region is anomalously warm? If so, why and what causes this seesaw pattern?
Response: By examining Figures 3 and 7, we speculate that this phenomenon relates to the position of the polar vortex displacement. When the polar vortex shifts or splits, causing it to skew towards only one region, that area experiences anomalously cold conditions. In the opposite direction of the displaced vortex, warm temperatures develop. This section describes the t2m and 500-hPa height anomalies. Most of the time, they are consistent. (L310-324)

L314-317: First of all, the differences between ERA5 and the model are not minor, especially from day -40 to day -10. The t2m anomalies are almost zero in this time period in the model. This again goes back to my previous concern about the robustness of the results in ERA5 and brings to the next question: do the t2m anomalies before SSWs precondition the cold anomalies after SSWs? This analysis cannot really show or support the role or importance of the t2m anomalies before SSWs in developing BOTH DWs. The same questions for other types of DWs.
Response: The simulated surface anomalies following the SSW onset are generally smaller in models. This is a common feature for models in existing literature. Although the composite amplitude in models is weaker than in ERA5, the general pattern is still similar. Further, the model results have also passed the significance test.

To well address your concern, we also made corresponding revision:
·  "In both datasets, for the type BOTH, cold anomalies develop following the SSW occurrence period from day 0 to day 40 over all the four regions (Europe, Asia, US, and Canada) …" (L334-337)

L322-323: Does the evolution of t2m here really relate to or result from SSWs for regions of Europe and Asia, given that the anomalies are not statistically significant and the t2m anomalies are almost unchanged during the period in Europe?
Response: From Figure 7g, it can be seen that the 100 hPa low anomaly center (strengthened polar vortex) has moved from East Asia to North America before and

after NA, and the temperature in East Asia rise. The temperature changes in Eurasia are closely related to the displacement and movement of polar vortex during the SSW. We have also made corresponding revision:

· "Namely, cold air outbreaks increase in North America, while the anomalously cold state gradually recovers to normal in Asia" (L342)

L355-356: It seems like the high IPV center is consistently over Europe throughout the whole period. There is no clear movement of high IPV from the figure, and there is a sudden strengthening of the anomalies, which indicates the IPV does not seem to be conserved but is under external forcing. Thus, it cannot tell the source of the IPV based on the movement of high IPV.

Response: You are right. We made corresponding revision:

· "Significant negative IPV anomalies develop over Greenland, consistent with the local anomalous high (see Fig. 3e), which helps advect cold air to eastern North America. In contrast, there are two high IPV centers in Eurasia, one over Europe, and the other over Central Asia. The anomalously high IPV center over Europe has remained stationary and continues to intensify." (L373-377)

Figure 7 and the relevant description: After reading the whole description of Figure 7, it is good to see different patterns in different types of DWs and NDW, but it is still not clear to me what dynamical processes or what and how these features and patterns before the SSWs influence the development of different types of DWs. The analysis is just descriptive, not really in-depth enough to reveal the differences in the underlying dynamics.

Response: Thank you for your criticism. The wave driving for the DWs and NDW is still, which is better shown in the E-P flux, which is shown in Figures 8-10.

Further, we also made several revisions for the description of Figure 7:

· "The anomalous high is situated at 60°E, while the anomalous low is situated over Northeastern Asia and Europe, implying that the polar vortex is likewise skewed towards Eurasia." (L437-439)

· "For the three types of DWs, during this period, negative anomaly centres emerged over Eurasia (or East Asia or Europe), indicating that the polar vortex was more inclined to shift towards Eurasia. This may be one reason why cold anomalies occurred across Eurasia during this period for all three types of DWs." (L455-458)

· "Similarly, centres of negative anomalies are also observed at 100 hPa over the regions where cold anomalies develop in the three types of DWs." (L478-480)

L480-481: From the vectors in figure 8e1 and g1, they seem to be pointing downward over the high latitudes, which indicates the opposite of what the authors describe here. Besides, the convergence is very weak in the stratosphere for BOTH and NA. Are the anomalies here statistically significant?

Response: We have revised the description and added significance tests for Fig. 8, 9, and 10.

- "The upward propagation is beginning to weaken for all types of DWs." (L513)
- "The black line represents the 90% confidence level for the composite E-P flux divergence anomalies." (L509)

L486-487: How does the short lifetime of the wave forcing explain the lack of impacts on the surface?

Response: Compared to DW, NDW has two characteristics of wave dynamic actions: lower intensity and shorter duration. Analysis of the NAM index shows that around occurrence of SSWs, the stratospheric circulation variations gradually propagate downward. Strong events usually have a more significant downward impact than weak events (mainly NDWs). Stronger events are preceded by active wave pulses. BOTHs are driven by both wave 1 and wave 2. NDWs are driven by weakly strengthened wave 1 and 2. The details has been shown in Figure 11.

The relatively short lifetime of wave forcing has a weak accumulative effect on the stratospheric disturbance. We revised this sentence:
- "The relatively short lifetime and weak intensity of the wave forcing for the NDW likely explains the relatively weak intensity of the stratospheric disturbance and even lacking impact on the near surface in the later period." (L519-521)

L503-505: The description is not accurate. Only Figure 9g1 shows this. The differences between the model and ERA5 are quite dominant.

Response: Thank you for your comment. We have made revisions and rephrased it.
- "Wave forcing in midlatitudes is still present for DWs in ERA5, and the anomalous E-P flux convergence also persists in the lower stratosphere at midlatitudes (Fig. 9e-g). For model simulations, although the upward flux still lasts, the convergence of the E-P flux has rapidly declined, and NA even shows divergence of the flux." (L538-542)

L512-514: There is a divergence in the lower stratosphere for the NA DW. Could the author provide any explanation for this? How does this feature lead to or play a role in the NA surface impact? Please explain this.

Response: The E-P flux shows divergence in the troposphere and lower stratosphere. It implies that the wave 2 forcing from the troposphere and lower stratosphere is present. In the upper stratosphere, the wave 2 cancel part of the wave 1 effect. This sentence is correct. The following sentence was revised. (L550-555)

L514-515: I disagree. The differences between the model and ERA5 are quite dominant to me.

Response: The difference between EAR5 and the model is mainly reflected in the divergence of NA and NDW E-P fluxes in the early stage of SSW. All are revised. (L553-555)

L517: "Both ERA5 and model simulations" Again, I think the differences between the model and ERA5 are large. For example, the model does not show the EP flux convergence anomalies for EA.
Response: We have made modifications to this description. (L557, L562-563)

L517-518: The arrows of the vectors are hard to see. This is also the question for Figures 8 and 9. In fact, the authors did not mention whether there is any scaling when plotting the EP flux, which is usually the convention. The small magnitude of the vectors indicates the anomalies are small. Are the EP flux anomalies statistically significant?
Response: We have revised Fig. 8, 9, and 10. We have adjusted the density of the arrows to enhance clarity. The units for the vectors are specified in the figure captions. We have also conducted significance tests for the composite flux divergence anomalies.

The scale is also added at the foot of the figure.

Discussion for Figures 8-10: I can see the differences in the EP flux and EP flux divergence across the three types of DWs. However, I think there is a lack of in-depth analysis of how these differences influence/determine the types of DWs. For example, after reading the whole section, I still do not know how the anomalous upward propagation of wave-2 wave activity after SSWs leads to the coldness over EA. Are the displacement or split of the vortex more important, or the wave activity itself more important?
Response: We analyzed the effects of wave activity and found that the intensity and duration of wave forcing affect the SSW events in their duration and intensity. The SSW duration and intensity then affect the NAM structure and persistency. The analysis of planetary waves 1 and 2 mainly explains the wave driving forcing for the stratospheric circulation anomalies. The SSW effect on the near surface is mainly diagnosed from height - time evolution of NAM. We did not intend to say which is more important.

Actually, we have considered that we might mislead our readers. We finally provided Figure 11 in the last round revision, which give the complete physical chain from wave forcing => SSW diversity => diverse NAM structure => different surface response.

Figure 11 and its summary: It is clear to summarize the main differences using this schematic. However, there is a lack of connection between all these dynamical processes and wave activities flux, and coupling, which lead to or contribute to the final development of different types of DWs. There is no clear interpretation to put all these components into a thread, which makes the study purely descriptive and lacks in-depth insights.
Response: Figure 11 gives a complete picture of how the different wave driving produce different SSWs and NAM structure. Finally, those differences are reflected in the surface response. This thread is clearly shown in Figure 11.
To well address your concern, we made future revision:

- "In the pre-onset period, across all three types of DWs, the anomalously low centres are present over Eurasia, which signifies that following the collapse of the polar vortex, its remnant tends to move towards Eurasia. For BOTH, the anomalously low band stretching from the North Atlantic to North America represents the portion of the polar vortex that also extends towards the America. In the post-onset period, anomalously low centers exist above each cold anomaly region. This indicates that around the time of DWs, the polar vortex either shifts or splits, and the movement of the collapsed polar vortex is a significant factor in the region experiencing cold anomalies." (L614-621)

L599-600: I don't see any improved understanding of the mechanisms. Please see my comments above.
Response: The text has been revised. Thank you for your strict criticism.

**Minor comments**:
L155 and Table 2: It will help to summarize the percentage of each type of event for both ERA5 and the model.
Response: The proportions of each event have been added. (L186-188)

L164: It would help to briefly describe the method by Maycock and Hitchcock (2015) so that it would be clearer when you mention the parameters you modified below.
Response: A brief description has been added. (174-181)

L238: What is Fig. 1e1 and f1?
Response: Removed.

L240: I agree there is nice consistency between the model and ERA5 in most cases here. However, the magnitude of the negative NAM is apparently weaker from the lower stratosphere to the mid-troposphere in the model than in ERA5. Something like this can be mentioned in the section where you introduce the overall model performance in downward impacts.
Response: Now added.
- "Physical quantities in the CESM2-WACCM model tend to exhibit weaker amplitudes than those in ERA5, although it simulates the downward influence of the SSW effectively, successfully reproducing general features." (Revised L581-583)

L249: I don't think the authors introduced how to compute or obtain the NAO index before.
Response: Now added.
- "After the NAM index is obtained at each pressure level, the index at 1000 hPa is used to represent the NAO index." (L142-144)

L254-255: How can DWs modify the PDF of the NAO?

Response: This sentence is changed.
- "···, which indicates that DWs have a significant effect on the surface circulation by projecting onto more negative NAO." (L271-272)

L291-292: Keep in mind that the t2m anomaly pattern contrastingly different for different types of events is not something new because this is how you define the three types of DWs and NDW.
Response: We agree with your viewpoint. In this sentence, we also emphasize the result constructed by definition.

L329-330: Figure 4b2 does not fully support this statement for Europe.
Response: "Eurasia" has been revised to "Asia." (L349)

L331-332: The Europe region in Figure 4c2 does not support the description.
Response: "Eurasia" has been revised to "Asia." (L351)

L333-334: This sentence is confusing. I don't understand which features or lines in the figure the authors talked about.
Response: We are describing the NDWs in Fig. 4d1 and d2. We have made the corrections. (L354)

L442: Isn't the 100 hPa height anomaly over the North Atlantic also negative?
Response: Yes, added (L468)

L525-529: Isn't the finding opposite to the previous studies?
Response: No, the results are consistent with previous studies. Refer to Figures 3. This study emphasizes the SSW diversity in influencing the near surface.